# Structure of Trypanosoma peroxisomal import complex unveils conformational heterogeneity

Ravi R. Sonani [1,4], Artur Blat[1,2,6], Malgorzata Jemiola-Rzeminska [3,6], Oskar Lipinski [1,2,5,6], Stuti N. Patel [1,6], Tabassum Sood [1,2,6] & Grzegorz Dubin [1] ✉

Peroxisomes are membrane enclosed organelles hosting diverse metabolic processes in eukaryotic cells. Having no protein synthetic abilities, peroxisomes import all required enzymes from the cytosol through a peroxin (Pex) import system. Peroxisome targeting sequence 1 (PTS1)-tagged cargo is recognized by cytosolic receptor, Pex5. The cargo-Pex5 complex docks at the peroxisomal membrane translocon, composed of Pex14 and Pex13, facilitating translocation into the peroxisomal lumen. Despite its significance, the structural basis of the process is only partially understood. Here, we characterize the cargo-Pex5-Pex14$_{NTD}$ ternary complex from *Trypanosoma cruzi*. Cryo-electron microscopy maps enabled model building for Pex5 (residues 327–462 and 487–653) bound to malate dehydrogenase (MDH; residues 1–323) cargo tetramer and Pex14$_{NTD}$ (residues 21–85). The model provides insight into conformational heterogeneity and identifies secondary interfaces. Specifically, we observe that orientations of Pex5 relative to MDH span a 17° angle. Additionally, PTS1- and Wxxx(F/Y)-independent contact surfaces are observed at MDH·Pex5 and Pex5·Pex14$_{NTD}$ interfaces, respectively. Mutational analysis indicates that the non-PTS1 MDH·Pex5 interface does not significantly contribute to the affinity, but limits the conformational heterogeneity of MDH-Pex5 interface. The Pex5·Pex14$_{NTD}$ interface constitutes an extended binding site of Pex14$_{NTD}$ over Pex5. We discuss the implications of these findings for understanding peroxisomal import mechanism.

Peroxisomes compartmentalize a set of important metabolic processes in eukaryotic cells, including catabolism of fatty acids and polyamines, biosynthesis of ether phospholipids, glyoxylate cycle in germinating seeds, glycolysis in *Trypanosoma* and many others[1,2]. Despite lacking genetic material and protein synthesis machinery, peroxisomes efficiently perform their functions by importing enzymes (referred to as cargo) from the cytoplasm[3]. The import process is orchestrated by a complex network of peroxin (Pex) proteins, numbering over a dozen. Pex proteins recognize specific peroxisomal targeting signal (PTS) sequences present on the folded cargo proteins in

[1]Malopolska Centre of Biotechnology, Jagiellonian University, Krakow, Poland. [2]Doctoral School of Exact and Natural Sciences, Jagiellonian University, Krakow, Poland. [3]Department of Plant Physiology and Biochemistry, Faculty of Biochemistry, Biophysics and Biotechnology, Jagiellonian University, Krakow, Poland. [4]Present address: Department of Biochemistry and Molecular Genetics, University of Virginia School of Medicine, Charlottesville, VA, USA. [5]Present address: Universite Claude Bernard Lyon 1, CNRS, Tissue Biology and Therapeutic Engineering Laboratory (LBTI), UMR 5305, Lyon, France. [6]These authors contributed equally: Artur Blat, Malgorzata Jemiola-Rzeminska, Oskar Lipinski, Stuti N. Patel, Tabassum Sood. ✉e-mail: grzegorz.dubin@uj.edu.pl

the cytoplasm. Subsequently, the cargo is translocated into the peroxisomal lumen, and the Pex system is recycled for a subsequent round of cargo import[4]. Remarkably, this translocation process occurs without unfolding of the cargo proteins[5,6].

The initial step of peroxisomal import involves recognition of a more common PTS1 sequence by soluble receptor Pex5 in the cytoplasm. The recognition of the less common PTS2 sequence requires an additional receptor, Pex7. Pex5 consists of two structural domains: N-terminal unstructured region guiding peroxisomal membrane translocation and C-terminal globular domain adopting tetratricopeptide repeat (TPR) fold and responsible for cargo recognition[7,8]. The PTS1 signal consists of variants of a Ser-Lys-Leu-COOH consensus sequence at the C-terminus of cargos[4]. Once bound by Pex5 in the cytosol, the cargo-Pex5 complex is subsequently translocated across the membrane by Pex13, and Pex14 assists in the process[9], but the structural details of the interactions and the mechanism of translocation are insufficiently understood.

The primary interaction between cargo and Pex5 is well-characterized through structures of isolated PTS1-peptide binding to the C-terminal TPR domain of Pex5[7,8]. The PTS1 peptide burrows into the central cavity of the Pex5 TPR domain[8]. Three structures of cargo-bound Pex5 TPR domain are available, which demonstrate secondary, PTS1-independent interactions between the cargo and Pex5 TPR domain. The extent to which secondary interactions contribute to cargo-Pex5 affinity varies significantly, and thus the role of secondary interactions is yet to be completely understood[10–14].

The process of translocating the Pex5-cargo complex across the peroxisomal membrane is subject to ongoing debate. What is established is that the import of the Pex5-cargo complex involves assistance from membrane-spanning proteins, Pex14 and Pex13[9]. It has been suggested that Pex5-cargo docking at the membrane is assisted primarily via interaction with Pex14. Different studies propose that cargo-Pex5, in conjunction with Pex14 and Pex13, facilitates the formation of a dynamic pore, or engages a preformed Pex13 pore, enabling translocation of the cargo[15–17]. N-terminal domain of Pex14 (Pex14_{NTD}) was additionally implicated in cargo extraction from the translocation pore. The nature and dynamics of the pore and the exact sequence of events require further investigation.

Pex14 consists of three structural regions: N-terminal Pex5-binding domain facing the peroxisomal matrix (Pex14_{NTD}), cytoplasmic coiled coil domain and C-terminal unstructured region. The interaction between Pex5 and Pex14 is primarily mediated through the recognition of multiple Wxxx(F/Y) motifs on Pex5 by Pex14_{NTD}[18–21]. This interaction has been extensively studied, typically through a system of short peptides containing the Pex5 Wxxx(F/Y) motif binding to Pex14_{NTD}[22]. However, a comprehensive understanding of the Pex14 binding site on Pex5 is still elusive, as structural data of full-length Pex5 (or extended TPR domain including at least one Wxxx(F/Y) motif) bound to Pex14 are currently unavailable. Earlier studies proposed that the binding of Pex14_{NTD} to the Pex5-cargo complex triggers cargo release[23], but the cooperative role of multiple copies of Pex14 at the Wxxx(F/Y) motifs within Pex5 in binding and/or release of the cargo remains uncharacterized[18–21].

In this study, we present the structure of the ternary complex involving cargo-Pex5-Pex14_{NTD} from *Trypanosoma cruzi*, the pathogen responsible for human Chagas disease[24]. Our structural analysis unravels the conformational heterogeneity of interactions of Pex5 with its cargo, and secondary interfaces between cargo-Pex5 and Pex5-Pex14_{NTD}. The role of the secondary interfaces is investigated by structure-guided mutational analysis. Furthermore, our results show that the Pex14_{NTD} binding at the Pex5 TPR domain proximal Wxxx(F/Y) site is not sufficient for structural reorganization of the Pex5 TPR domain and cargo release in vitro.

## Results

### Reconstitution of cargo-Pex5-Pex14_{NTD} ternary complex

Specialized peroxisomes, known as glycosomes, import glycolysis-associated enzymes (cargo) from the cytosol through the Pex import system and compartmentalize glycolytic reactions in *Trypanosoma*. The Pex import process is a validated drug target in Chagas disease[25–27]. After assessing various cargos (glycolytic enzymes) and Pex constructs (Supplementary Table 1), we chose glycosomal malate dehydrogenase (MDH, residues 1–323, UniProt Q4DRD8) as a cargo, and full length Pex5 (residues 1–666, GenBank PBJ69826.1) and N-terminal domain of Pex14 (Pex14_{NTD}; residues 21–85, GenBank RNC55913.1) for in vitro reconstitution of the ternary complex. MDH is a crucial enzyme for the glycolysis auxiliary pathway, maintaining the pool of malate/oxaloacetate[28,29]. All three proteins were individually expressed in *Escherichia coli* and purified to homogeneity. The complex was reconstituted in vitro by mixing purified proteins and separating the complex from non-complexed components using size exclusion chromatography (SEC). The SEC profile of mixed components showed additional peaks at a higher molecular weight, corresponding to the ternary complex (Supplementary Fig. 1). The presence of all three proteins in these peaks was confirmed by the SDS-PAGE (Supplementary Fig. 1).

### Cryo-EM reconstructions and heterogeneity of the ternary complex

Upon initial cryo-EM data analysis, the tetrameric arrangement of MDH was evident, yet significant variability in the MDH:Pex5 stoichiometry was noted (Fig. 1). Avoiding chemical crosslinking, which could potentially homogenize the population, we instead extracted information on compositional heterogeneity from the available dataset. Classification efforts yielded 2D class averages representing diverse compositions of complexes, prominently featuring densities corresponding to complexes with one Pex5 molecule (MP1), two Pex5 molecules (MP2), and three (or four; the exact stoichiometry is ambiguous in 2D class average due to the dihedral symmetry) Pex5 molecules (MP3/4) bound to the MDH tetramer (Fig. 1A). The density corresponding to the third Pex5 molecule in MP3/4 was relatively weak (Fig. 1A).

We first attempted to reconstruct high-resolution 3D maps of MP1, MP2 and MP3/4 using relevant particles and data processing procedures not accounting for conformational variability (see Methods). We successfully reconstructed MP1 and MP2 maps showing one and two copies of Pex5 bound to MDH tetramer, respectively (Fig. 1B). Processing of MP3/4 particles did not produce a reasonable quality map, likely due to the low occupancy of the 3rd (and/or 4th) copies of Pex5 and was not analyzed further.

MP1 and MP2 maps show a well-resolved MDH region, but poorly resolved Pex5 region(s), indicating conformational heterogeneity in the latter component (Fig. 1). The resulting maps facilitated de novo model building for MDH, forming a tetramer of ~10 nm diameter. The tetramer is composed of two dimers related by dihedral (D2) symmetry (Fig. 1C). The peroxisomal targeting signal 1 (PTS1) motifs of all MDH subunits are found at the periphery of the tetramer and are equally accessible to the solvent (Fig. 1C).

To elucidate the conformational variability and improve the density of the Pex5 region, we employed three-dimensional variability analysis (3DVA) using cryoSPARC[30]. This approach improved the density of the Pex5 region and revealed significant heterogeneity in Pex5 orientations relative to MDH, as depicted in Supplementary Movies 1 and 2. The resulting data facilitated reconstruction of distinct 3D volumes representing various orientations of Pex5 relative to MDH within the MP1 and MP2 complexes. We refined and built two representative atomic structures for each complex state, capturing two

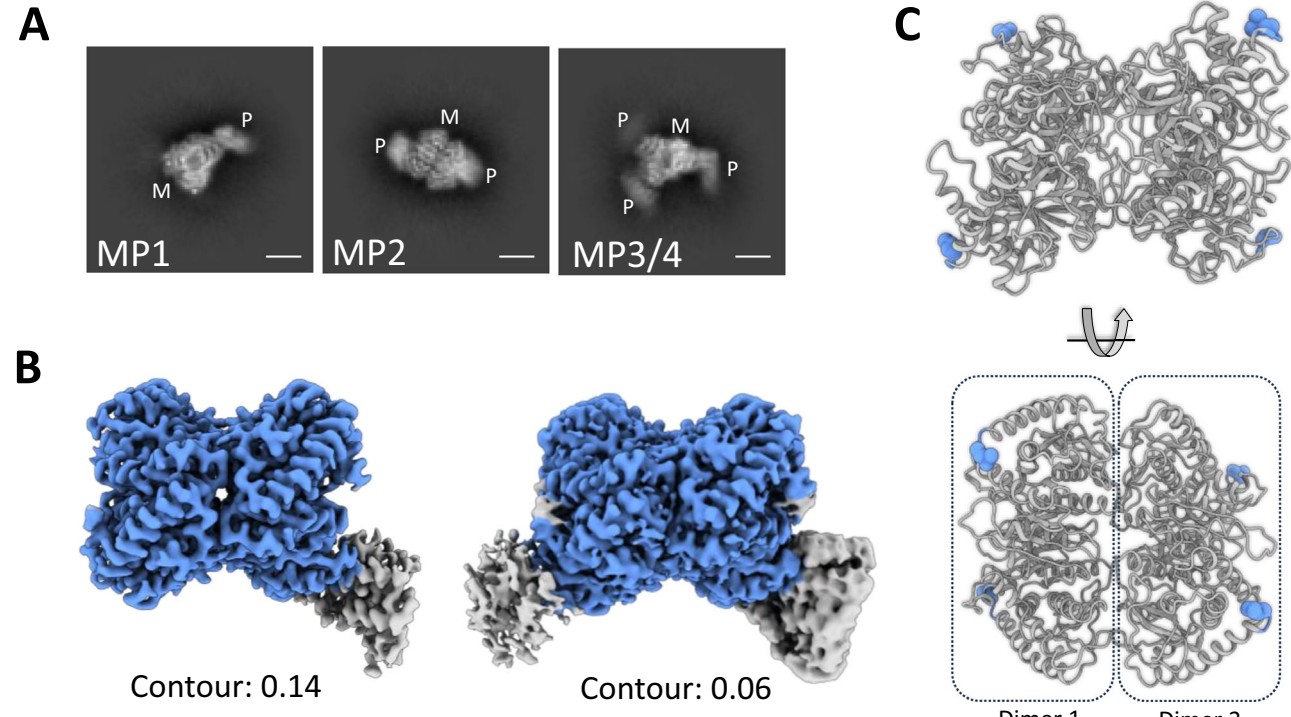

**Fig. 1 | Cryo-EM analysis of peroxisomal import ternary complex—MDH·Pex5·Pex14_NTD through processing pipeline not accounting for conformational variability (see Methods). A** Representative 2D class averages (from total of 100 class averages) showing density corresponding to the MDH-tetramer (M) and Pex5 (P). Three compositionally different complexes—MP1, MP2, and MP3/4 were observed showing one, two, and three (or four) copies of Pex5 bound to the MDH-tetramer, respectively. Scalebar, -50 Å. **B** 3D density maps of MP1 (left; contour level 0.14) and MP2 (right; contour level 0.06) complexes show nicely resolved density for the MDH tetramer (blue density), but blurred density for the bound Pex5 (grey density). **C** The MDH tetramer structure shows equal accessibility of all four PTS1 motifs (blue balls), suggesting that no steric occlusion is present at any of these sites, which could preclude Pex5 binding. Two MDH-dimers related by dihedral symmetry are indicated by dotted rectangles.

extreme conformations, namely, MP1-close (MP1-c), MP1-distal (MP1-d), MP2-close (MP2-c), and MP2-distal (MP2-d) (Fig. 2; Supplementary Fig. 2 and Supplementary Movies 3–7). The refined maps facilitated model building of MDH and Pex5 residues 327–462 and 487–653, encompassing the TPR domain and an upstream helix containing the Wxxx(F/Y) motif (residues 327–345) (Supplementary Figs. 3–6). The helix distinguishes our structure from previously available Pex5 TPR domain crystal structures[8], in which the Wxxx(F/Y) motif-containing helix is not resolved. The density corresponding to the rest of Pex5 N-terminal domain (NTD; residues 1–326) was not discernible in our structure, indicating its flexible nature even in the presence of Pex14_NTD, consistent with observations from prior studies[31].

It is of note that while we observed particles with two Pex5 molecules each bound to the different dimer of the MDH tetramer, we did not observe particles with two Pex5 molecules bound to the same dimer (Fig. 2C). The underlying reason for this specific topology remains unclear as the PTS1 binding sites on all protomers are equally accessible (Fig. 1C). Furthermore, the presence of particles representing the MP3/4 complex (Fig. 1A), where three or four Pex5 molecules are bound to MDH tetramer (two Pex5 on one MDH-dimer and one or two Pex5 on another MDH-dimer), suggests that there are no steric clashes between two Pex5 molecules bound to single MDH-dimer. However, it is uncertain whether the binding of more than one Pex5 is necessary for the import of the oligomeric cargo. The piggy-backing of cargo isoforms lacking the PTS targeting signal within heteromultimeric complexes suggests that the binding of Pex5 to all subunits of the oligomer is not required for translocation[5,6]. Furthermore, recent findings indicate that predominantly a single Pex5 molecule binds to Eci1 hexameric cargo, implying that the binding of a single Pex5 molecule is sufficient for efficient import[14].

## Pex5-binding on MDH is characterized by conformational heterogeneity

Recognition of the MDH by Pex5 is mediated via PTS1 motif at the central cavity of Pex5 TPR domain (Fig. 2B). The interaction is stabilized by a network of hydrogen bonds involving the PTS1 C-terminal carboxylate group, mainchain acceptors, and lysine sidechain amine, as observed previously in different species[8] (Supplementary Fig. 7). Some rearrangement of this network is evident between the distal and close states. For instance, the side chain of MDH(K322) forms an H-bond with Pex5(E407) in the distal state, while in the close state, it forms an H-bond with Pex5(N439) (Supplementary Fig. 7). Similarly, H-bonds involving the C-terminus of MDH(L323) are partly rearranged between the distal and close states (Supplementary Fig. 7).

The overall geometry of the PTS1-bound Pex5 TPR domain in *T. cruzi* is compact compared to the structure of the apo-Pex5 TPR domain[32]. Similar compaction was also observed for *Trypanosoma brucei* and the human TPR domain of Pex5[7,8,10,12,32], suggesting that TPR domain compaction upon PTS1 binding is a general phenomenon across different organisms.

The 3D structural variability (3DVA) analysis reveals that Pex5, when bound to PTS1, exhibits conformation variability on the surface of MDH. We categorized the raw particles into 20 clusters, each representing varying tilt covering the range of orientations between distal and close conformations. The 3DVA results reveal the presence of particles across the whole range of the conformational space between close and distal states (Supplementary Fig. 2). This conformational heterogeneity can be characterized by two components: tilt and twist. Pex5 TPR domain tilts nearly as a rigid body between close and distal conformations with a maximum tilt angle of 17° relative to the PTS1 (Fig. 2A; and Supplementary Movies 1–7). Additionally,

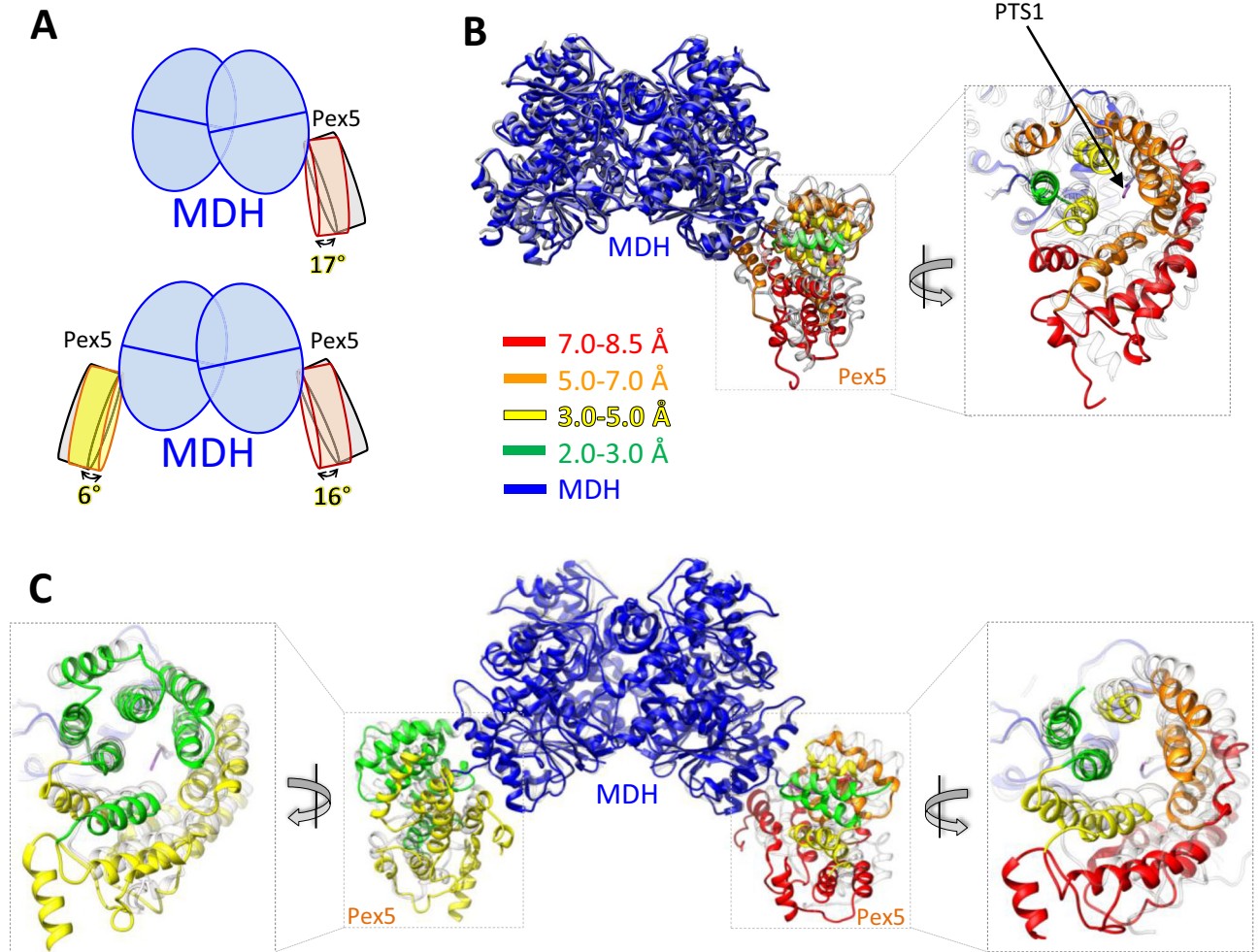

**Fig. 2 | The Pex5-cargo interface shows significant conformational hetero-geneity. A** Schematic representation of angular tilt of TPR domain of Pex5 relative to the MDH in MP1 (upper panel) and MP2 (lower panel) structures. Superimposed 3D structures of close (colored) and distal (grey) conformations of MP1 (**B**) and MP2 (**C**) complex. Color codes in close conformation represent the extent of tilt between distal and close confirmations as explained in the inset. MDH is colored blue in all panels. Pex5 TPR domain is colored according to the discussed features. Refer to Supplementary Fig. 8 for residue-wise RMSD between distal and close orientations.

Pex5 TPR domain twists up to 13° relative to the axis roughly parallel to the TPR3 motif (Supplementary Movie 8). The combined tilting and twisting result in a radial displacement of Pex5 TPR domain ranging from 3 to 8 Å (Fig. 2). Specifically, TPR motifs 1, 2, 6, and 7 (residues 330–425, 527–640) show displacements of 5–8 Å, which is greater than the displacement of the central TPR motifs 3, 4, and 5 (residues 428–525) that is ~3.5 Å (Figs. 2B, C and Supplementary Movies 1–8). Further, 3DVA results suggest that there is no coordination between the variability of two Pex5 molecules in the MP2 complex (Supplementary Fig. 9).

**Secondary MDH-Pex5 interfaces do not significantly contribute to affinity**

Along PTS1-mediated interaction, two additional interfaces between MDH and Pex5 TPR domain are identified by PISA analysis (Fig. 3A). Interface 1 involves MDH residues 62–70 and Pex5 residues 439–444, forming a loop connecting α6 and α7. Notably, MDH residues 62–70 form a loop (MDH$_{62-70}$) unique to the glycosomal isoform and absent in the cytosolic/mitochondrial isoforms and homologs from other organisms (Supplementary Fig. 10), suggesting its possible specific role in glycosomal import[33]. Interface 2 involves MDH residues 142–145 and Pex5 helix α15 made up of residues 625–633 (Fig. 3A).

Because close analysis of interfaces 1 and 2 does not reveal clas-sical features of protein-protein interaction[34] and because identified conformational heterogeneity affects the extent of the interfaces, we investigated if the identified interfaces contribute to affinity. Mutants were generated targeting residues at the MDH-Pex5 interface, and their affinities were determined by isothermal titration calorimetry (ITC) (Table 1). The mutants were evaluated in the context of Pex5$_{eTPR}$ (residues 314–666; Supplementary Table 4), a truncated Pex5 con-struct slightly exceeding the region defined by density in our structure (residues 327–462 and 487–653). The affinity of Pex5$_{eTPR}$ and MDH was estimated at $K_d$ ~ 435 nM (Table 1). To disrupt interface 1, we replaced the MDH$_{62-70}$ loop with a GS-linker, resulting in MDH(Δ62–70). This resulted in a moderate decrease (~2.3-fold) ($p < 0.001$) in affinity towards Pex5$_{eTPR}$, indicating little contribution of the interface to affinity. Interface 1 at the Pex5 side is composed of backbone atoms and thus not directly amenable to mutational analysis. To assess the contribution of interface 2, a double mutant of Pex5$_{eTPR}$(R625A, D629A) was constructed. The mutant exhibited an affinity comparable to the wild type Pex5$_{eTPR}$ ($p = 0.5$), indicating insignificant contribution of interface 2 to MDH-Pex5 affinity. The MDH side of the interface was composed of backbone atoms and thus was not analyzed by muta-genesis. To further explore the collective impact of secondary MDH-

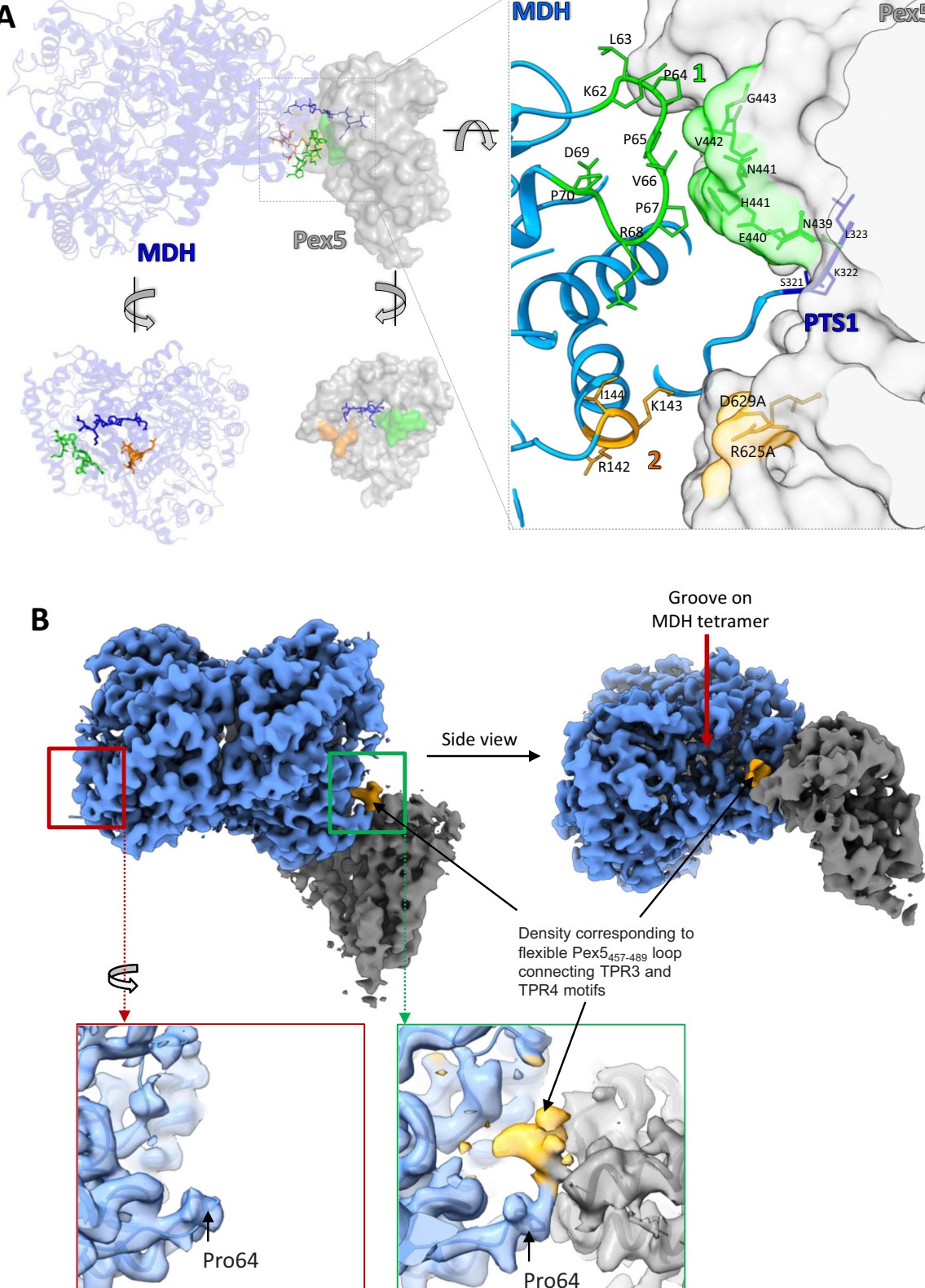

**Fig. 3 | Non-PTS1 interactions of the Pex5 TPR domain and MDH. A** Non-PTS1 interactions are represented by green and orange colors on the MP1-d structure (upper panel), MDH only (lower left panel) and Pex5 only (lower right panel). MDH and Pex5 are depicted in blue ribbon and grey surface, respectively. PTS1 burrowing inside the TPR-domain cavity is presented by the blue-stick model. Right panel shows the enlarged view of MDH·Pex5 PTS1 involving interface and non-PTS1 interactions. See Supplementary Fig. 12 for PTS1 and non-PTS1 interactions in close state. **B** Density corresponding to the Pex5$_{457-489}$ loop transiently interacting with the MDH groove in MP1-d map (gold). The density is not seen in a groove at the Pex5 unoccupied site (red square). See Supplementary Fig. 11 for analogous analysis of MP2-d map.

**Table 1 | Mutational analysis of MDH-Pex5$_{eTPR}$ interface**

| Pex5$_{eTPR}$ | MDH | $K_d$ [nM] | $\Delta H$ [kJ/mol] | $-T\Delta S$ [kJ/mol] | $\Delta G$ [kJ/mol] | N[a] | Statistical significance (p value) |
|---|---|---|---|---|---|---|---|
| Wild type | Wild type | 435 ± 80 | −26.5 ± 0.8 | −9.8 ± 3.5 | −36.3 ± 0.2 | 0.398 ± 0.007 | - |
| Wild type | GSGS[b] | 148 ± 26 | −24.0 ± 0.4 | −15.0 ± 1.8 | −39.0 ± 0.3 | 0.543 ± 0.005 | <0.001 |
| | Δ(62–70)[c,d] | 1001 ± 121 | −25.4 ± 0.7 | −8.8 ± 1.8 | −34.3 ± 0.2 | 0.500 ± 0.009 | <0.001 |
| | PTS1[e] | 135 ± 19 | −21.0 ± 0.2 | −18.2 ± 0.7 | −39.2 ± 0.3 | 0.933 ± 0.006 | <0.001 |
| R625A, D629A[f] | Wild type | 453 ± 64 | −22.5 ± 0.5 | −13.7 ± 1.0 | −36.2 ± 0.2 | 0.500 ± 0.006 | 0.5 |
| (Δ470–480) | | 102 ± 19 | −28.8 ± 0.5 | −11.1 ± 2.2 | −40.0 ± 0.5 | 0.452 ± 0.004 | <0.001 |
| P490R | | 1389 ± 226 | −26.9 ± 0.9 | −6.0 ± 2.3 | −32.9 ± 0.5 | 0.292 ± 0.006 | <0.001 |
| (Δ470–480) | PTS1[e] | 420 ± 93 | −26.4 ± 0.6 | −10.0 ± 2.8 | −36.4 ± 0.3 | 1.040 ± 0.015 | <0.001 |

Interactions of indicated mutants were characterized by Isothermal Titration Calorimetry (ITC). Welch's t-test (two-tailed) p value is reported for comparing $K_d$ values. Variants are compared to wild type; (Δ470–480)/PTS1 is compared to WT/PTS1.
[a]N value is derived from data fitting and is related to binding stoichiometry with following formula: N = St*(AF$_{cell}$/AF$_{syr}$), where St—stoichiometry, AF$_{cell}$—active fraction of protein in the cell (MDH), AF$_{syr}$—active fraction of protein in the syringe (Pex5$_{eTPR}$), [b]disrupted sites 1 and 2, [c]disrupted site 1, [d]62-KLPPVPRDP-70 loop was substituted with GS linker, [e]PTS1 peptide derived from MDH (NH2-ARSKL-COOH), [f]disrupted site 2.

Pex5 interfaces, a linker (GSGS) was introduced between PTS1 and MDH to physically separate the binding partners, thus disrupting all secondary interactions. The MDH(GSGS) linker variant demonstrated nearly three times higher affinity compared to the wild type MDH ($p < 0.001$) (Table 1). The change in affinity was entropy-driven, suggesting that the secondary interactions have an overall negative effect on affinity by restricting the conformational heterogeneity of the wild-type interface. The affinity of MDH(GSGS) is comparable to that of a short peptide encompassing PTS1 of MDH (residues 319-ARSKL-323; $p > 0.1$) (Table 1), again suggesting that the introduction of GSGS linker indeed abolished all secondary interactions at the Pex5-MDH interface.

The above findings collectively demonstrate that neither of the identified secondary interfaces significantly contributes to the MDH-Pex5$_{eTPR}$ affinity and only restrict the conformational heterogeneity of Pex5 TPR domain binding at the surface of MDH.

**Transient interaction of Pex5$_{457-489}$ loop with MDH**
The maps accounting for the distal states of MDH-Pex5 complexes (MP1-d and MP2-d) show additional density next to Pex5 residue Ser491 and facing MDH$_{62-70}$ loop (Fig. 3B; Supplementary Fig. 11). Notably, the Pex5$_{457-489}$ loop (linking TPR motifs 3 and 4) is unresolved in all existing crystal structures, indicating its flexible nature. The additional density is observed only at sites occupied by Pex5, and not at the vacant PTS1 sites, indicating that it corresponds to a fragment of the Pex5$_{457-489}$ loop (Fig. 3B; Supplementary Fig. 11). Additionally, the density was absent in the map of MDH tetramer solved in this study (Supplementary Fig. 11), further suggesting it originates from Pex5. These results indicate restriction of conformational flexibility of the Pex5$_{457-489}$ loop in the distal state of the MDH-Pex5 complex.

The identified additional density was insufficient to construct an atomic model of the Pex5$_{457-489}$ loop, but indicates that the loop extends into the groove between MDH subunits (Fig. 3B). In glycosomal MDH, the groove is notably deeper (volume ~4900 Å$^3$) compared to non-glycosomal MDH (PDB id: 5ZI4) (volume ~2500 Å$^3$) owing to the insertion of MDH$_{62-70}$ loop, characteristic only for the glycosomal MDH. To determine the significance of the identified interaction, we tested the binding of Pex5 mutant with a partial deletion of the loop [Pex5$_{eTPR}$(Δ470–480)] with MDH. Pex5$_{eTPR}$(Δ470–480) retained the ability to recognize MDH-derived PTS1 peptide, indicating the deletion had no significant impact on the structure of the mutant, though the affinity was moderately (~3.1×) decreased compared to the wild type ($p < 0.001$; Table 1). At the same time, the mutant exhibited more than four times higher affinity for MDH compared to wild-type Pex5$_{eTPR}$ ($p < 0.001$). This suggests that the Pex5$_{470-480}$ loop sterically clashes with MDH in the distal state of the complex, limiting the available tilt and affinity of the interaction. To further test the hypothesis, we

mutated the Pro490 residue, which breaks the TPR4 helix at the onset of the Pex5$_{457-489}$ loop into arginine, a residue which we expected to support the expansion of the TPR4 helix and thus limit the loop flexibility. The affinity of Pex5$_{eTPR}$(P490R) mutant was compromised more than 3-fold compared to the wild type ($p < 0.001$), and the change in affinity was primarily driven by the entropic component, again demonstrating that the interactions of Pex5$_{457-489}$ loop at the inter-subunit groove of MDH limit the conformational variability available for the complex.

**Pex14$_{NTD}$ binding site and Wxxx(F/Y) independent interactions with Pex5**
We observed weak density beyond the Pex5 TPR domain, which was difficult to interpret in any of the 3DVA-derived maps and was not examined for the purpose of variability analysis described in prior paragraphs. The density could have corresponded to either a fragment of the Pex5 N-terminal domain or Pex14$_{NTD}$. To clarify the ambiguity, we reconstituted in vitro and solved the structure of the MDH-Pex5 binary complex devoid of Pex14$_{NTD}$ (see Methods). The map for the binary complex revealed no density beyond the MDH and Pex5 TPR domain (Supplementary Fig. 13), indicating that the additional density observed in the ternary complex originates from Pex14$_{NTD}$. In the ternary complex map, we improved this density by focused refinement using all collected data. Such data treatment did not account for the conformational heterogeneity of the MDH-Pex5 interface, but resolved the density features in the Pex region and enabled the unambiguous atomic model building of Pex14$_{NTD}$ located on the Pex5 TPR proximal Wxxx(F/Y) motif. The final atomic model of the ternary complex (designated as MP1P) was built in a composite map comprising the consensus MDH density map derived as described earlier in the text (Fig. 1) and the Pex5-Pex14$_{NTD}$ density map derived from focused refinement. The MP1P model consists of the MDH tetramer, Pex5 (residues 327–462, 487–653), and Pex14$_{NTD}$ (residues 21–85) (Fig. 4; Supplementary Fig. 14).

The ternary complex structure constructed in MP1P composite map reveals an extended Pex14$_{NTD}$-Pex5 binding site involving two distinct interfaces (Fig. 4). The first interface encompasses Pex5 Wxxx(F/Y) motif[22] (328-WVREY-332) constituting a part of an α-helix (residues 327–345) extending beyond the globular structure of the TPR domain of Pex5 (Fig. 4). The core interactions in this region are contributed by bulky sidechains within the 328-WVREY-332 motif (underlined) which bury in a hydrophobic groove on the surface of Pex14$_{NTD}$. The Wxxx(F/Y) interface resembles closely that observed in previously solved structures of short peptides encompassing the Wxxx(F/Y) motif in complex with Pex14$_{NTD}$[22] (Supplementary Fig. 15). The second, non-Wxxx(F/Y) interface has not been observed before

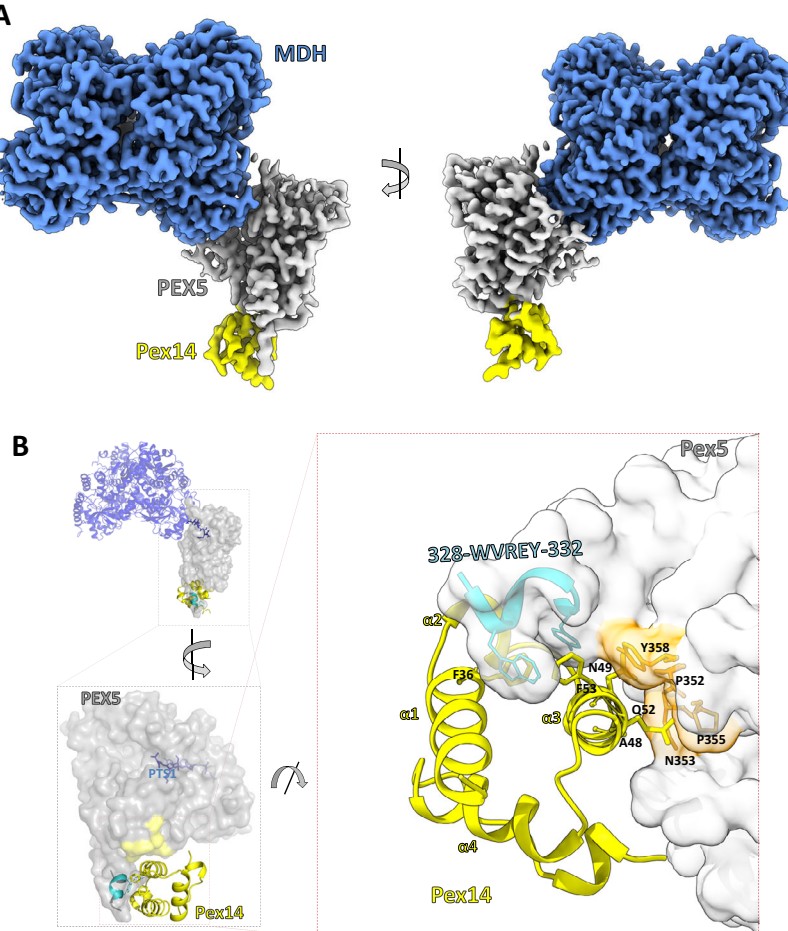

**Fig. 4 | Ternary complex structure (MP1P) and Pex5 – Pex14$_{NTD}$ interfaces.**
**A** Composite cryo-EM map of MDH·Pex5·PEX14$_{NTD}$ ternary complex (MP1P) demonstrating well resolved structural features of MDH, Pex5 (residues 327–462 and 487–653; Pex5 TPR domain containing the proximal Wxxx(F/Y) motif) and Pex14$_{NTD}$. **B** Details of Pex14$_{NTD}$ interaction with Pex5 as seen in a model interpreting the map presented in panel A. MDH, Pex5 and Pex14$_{NTD}$ are depicted in blue ribbon, grey surface and yellow ribbon, respectively. Wxxx(F/Y) motif is highlighted cyan. Non-Wxxx(F/Y) interaction region is highlighted orange. Helices of Pex14$_{NTD}$ are annotated in yellow. Residues most relevant for the interactions are shown in the stick model.

and involves Pex14$_{NTD}$ helix α3 and 353-NNPYMY-358 motif of Pex5 (Fig. 4 and Supplementary Figs. 16, 17). The non-Wxxx(F/Y) interface contributes ~800 Å² of buried surface area out of a total ~1300 Å² comprising the Pex14$_{NTD}$-Pex5 interface. The remaining ~500 Å² are contributed by the Wxxx(F/Y) site. Helix α3 of Pex14$_{NTD}$ (Supplementary Fig. 17) and two amino acids (underlined) in 353-NNPYMY-358 motif of Pex5 are conserved (Supplementary Fig. 16). The 353–NNPYMY-358 region forms an extended interhelix linker separated from surrounding helical regions by helix-breaking P352 and P355. The resolution limits detailed analysis of interactions, but CB atoms are clearly discernable, allowing to roughly predict the position of the sidechains. Sidechains of Pex5 N353 and Y358 extend in the direction of Pex14, but seem not to be involved in any specific interactions. The sidechains of A48 and Q52 of Pex14$_{NTD}$ extend in the direction of Pex5, but again, no specific interactions can be predicted in the region. Overall, the interface does not resemble protein-protein interaction interfaces[34], but rather a proximity assembly.

To evaluate the contribution of the non-Wxxx(F/Y) interface to Pex5-Pex14$_{NTD}$ affinity, we designed and tested the affinity of relevant mutants in the context of Pex5$_{eTPR}$. Affinities were determined in the presence of the PTS1 peptide to maintain a cargo-bound-like conformation of Pex5$_{eTPR}$, presumably a physiologically relevant binding conformation (affinities in the absence of PTS1 are presented for reference in Supplementary Table 2). The affinity ($K_d$) of the wild-type Pex14$_{NTD}$ for Pex5$_{eTPR}$ was estimated at ~53 nM by ITC (Table 2). The affinity of Pex14$_{NTD}$(Q52A) mutant was comparable ($p > 0.8$) to the wild type, indicating that Q52 does not contribute specific interactions with Pex5$_{eTPR}$. The affinity of Pex14$_{NTD}$(A48F) decreased more than two times compared to the wild type ($p < 0.001$). This demonstrates that the sidechain of A48 indeed points in the direction of Pex5$_{eTPR}$ as predicted, but the interface is flexible enough to accommodate a bulky sidechain in place of a short sidechain of Ala48 with only insignificant steric hindrance. Mutating Y358 of Pex5$_{eTPR}$ to alanine decreased the affinity of the mutant by ~1.5 times compared to the wild type, again as predicted in terms of involvement of Y358 in the interface, though the effect has not reached statistical significance ($p = 0.07$). Minor influence of Y358A residue substitution again suggests that the interface is flexible to accommodate significant changes in sidechain properties without a large effect on affinity. Pex5$_{eTPR}$(N353E) had almost no effect on affinity ($p > 0.4$), demonstrating that hydrogen bonding is not involved in the interaction. To further substantiate the above findings, we introduced a GSGS-spacer between the Pex5$_{TPR}$ domain and Wxxx(F/Y) motif to retain the primary interface, but physically separate the secondary interaction surface. The mutation resulted in a roughly 3.5-fold decrease in the affinity ($p < 0.001$), demonstrating that both the Wxxx(F/Y) and non-Wxxx(F/Y) sites contribute to the Pex14$_{NTD}$-Pex5$_{eTPR}$ affinity, with the former site having a major contribution.

**Table 2 | Mutational analysis of Pex14$_{NTD}$-Pex5$_{eTPR}$ interface**

| Pex5$_{eTPR}$ | Pex14$_{NTD}$ | $K_d$ [nM] | $\Delta H$ [kJ/mol] | $-T\Delta S$ [J/molK] | $\Delta G$ [kJ/mol] | N[a] | Statistical significance (p value) |
|---|---|---|---|---|---|---|---|
| Wild type | Wild type | 53 ± 12 | −64.5 ± 0.8 | 22.96 | −41.55 | 0.671 ± 0.005 | - |
| Wild type | Q52A | 54 ± 32 | −65.7 ± 2.4 | 24.21 | −41.49 | 0.664 ± 0.014 | 0.89 |
| | A48F | 121 ± 62 | −76.5 ± 3.3 | 37.00 | −39.47 | 0.608 ± 0.016 | <0.001 |
| Y358A | Wild type | 81 ± 55 | −57.4 ± 2.4 | 16.93 | −40.48 | 0.880 ± 0.022 | 0.07 |
| N353E | | 45 ± 39 | −57.8 ± 2.0 | 15.86 | −41.96 | 0.659 ± 0.011 | 0.45 |
| GSSG | | 178 ± 43 | −52.6 ± 1.2 | 14.04 | −38.53 | 0.700 ± 0.010 | <0.001 |

Interaction of indicated mutants was characterized by Isothermal Titration Calorimetry (ITC) in the presence of PTS1 peptide. The p value derived from the Welch's t-test (two-tailed) was used for comparing $K_d$ values. Variants are compared to wild type. [a]N value is derived from data fitting and is related to binding stoichiometry with the following formula: $N = St*(AF_{cell}/AF_{syr})$, where St—stoichiometry, $AF_{cell}$—active fraction of protein in the cell (Pex5$_{eTPR}$ + PTS1), $AF_{syr}$—active fraction of protein in the syringe (Pex14$_{NTD}$).

Structural rearrangement within the TPR domain of Pex5 was demonstrated upon PTS1 binding[12]. Since we demonstrated that Pex14$_{NTD}$ interacts directly with the TPR domain, we tested the potential influence of Pex14$_{NTD}$ on Pex5 affinity for PTS1. The affinity of the fluorescently labeled PTS1 peptide for Pex5$_{eTPR}$ was determined at $K_d = 254 \pm 9$ nM by fluorescence polarization. The affinity was ~3.2 times higher in presence of excess Pex14$_{NTD}$ ($K_d = 80 \pm 5$ nM; $p < 0.001$) (Supplementary Fig. 18). Comparable increase in affinity of full length Pex5 (containing three Wxxx(F/Y) motifs) for PTS1 in the presence of Pex14$_{NTD}$ was observed ($K_d = 88 \pm 5$ nM and $K_d = 37 \pm 2$ in the absence and presence of Pex14$_{NTD}$, respectively; $p < 0.001$). These data indicate that at the tested conditions, Pex14$_{NTD}$ binding does not release the cargo from Pex5 (or Pex5$_{eTPR}$). In contrast, it slightly increases the affinity of Pex5 (and consistently of Pex5$_{eTPR}$) for the cargo. In turn, the affinity of Pex14$_{NTD}$ for Pex5$_{eTPR}$ determined by ITC in the presence of PTS1 peptide ($K_d = 53 \pm 12$; Table 2) is lower ($p < 0.001$) compared to the affinity determined in the absence of PTS1 peptide ($K_d = 35 \pm 5$; Supplementary Table 2).

## Discussion

Earlier studies have shown that PTS1 guided cargo recognition by Pex5 exhibits both autonomous[13] (PTS1-mediated, cargo structure independent) and non-autonomous[11] (PTS1-independent, cargo structure dependent) characteristics. Non-autonomous model is exemplified by alanine-glyoxylate aminotransferase (AGT) which binding to Pex5 is substantially enhanced by secondary interactions[11]. In contrast, it has been demonstrated that the secondary interface does not significantly contribute to the affinity of Pex5 for human sterol carrier protein 2 (SCP2)[10,13]. Nanomolar affinity for Pex5 of short peptides encompassing PTS1 (in the absence of any secondary interactions) and successful import of PTS1-appended non-peroxisomal proteins into peroxisomes (in the absence of evolutionary optimized secondary interactions) support the predominant importance of the autonomous binding[35–38]. Our data align with the above results by indicating that the secondary interface contributes only weakly to Pex5$_{eTPR}$ affinity for MDH (~3x difference in affinity for Pex5 between WT and GSGS mutant of MDH; Table 1). At the same time, our data suggests that the secondary interactions modulate the conformational heterogeneity of the cargo-Pex5 complex.

Prior studies involving AGT and SCP2, while providing significant insights in cargo Pex5 interactions, have not accounted for the potential variability of the interface as crystal structures were analysed[11,13]. Our study demonstrates enhanced affinity of Pex5$_{eTPR}$ for MDH upon abolishing of the secondary interfaces (GSGS insertion mutant, Table 1). Similar effect was observed earlier for SCP2[13]. The effects are not pronounced, and we postulate that the secondary interactions (including the interactions of Pex5$_{457–489}$ loop) only provide proximity clashes, which limit the conformational heterogeneity rather than contribute to affinity. Following our findings in

*Trypanosoma*, a comparable conformational heterogeneity of the Pex5 TPR domain relative to cargo was identified in yeast[14]. The consensus structural variability component in cargo recognition in two distant organisms suggests that the characteristics may be common to peroxisomal import. Evolutionary adjustment of the interface region seen in peroxisomal MDH in *Trypanosoma* (Supplementary Fig. 10) suggests the physiological relevance of finetuning the conformational variability of the interaction. We speculate that the observed conformational variability plays a role in translocation through the peroxisomal membrane, and proximity assemblies provide a cargo-independent mechanism of limiting the variability, but further studies are needed to support such a hypothesis.

Our data demonstrates the conformational heterogeneity of MDH-Pex5 interactions, but the detailed description of conformational variability is limited by considerations of cryo-EM data analysis, including the necessity of substantial averaging to increase the signal-to-noise ratio, which limits the granularity of obtained models. The number of clusters and associated maps resulting from 3DVA analysis is shaped by input parameters and should not be interpreted as a particular number of discrete states. Presence of particles across the whole conformational space between the close and distal states of the complex in 3DVA (Supplementary Fig. 2) is consistent with a relatively low energy contribution of PTS1-independent interactions. However, the data do not distinguish between continuous, quasi-continuous or discrete distribution of Pex5 orientations relative to MDH. The most probable global model assumes equilibrium of short-lived states (ensembles) with a low-energy barrier between the states and spanning the space defined by a maximum tilt of 17° and twist of 13° of the TPR domain of Pex5 relative to MDH.

Prior studies thoroughly characterized the interaction of Pex14$_{NTD}$ with short peptides containing the primary recognition motif characterized by a consensus sequence of Wxxx(F/Y)[22], but the binding of Pex14$_{NTD}$ relative to the Pex5 TPR domain remained unclear. Our structure of the ternary complex provides direct visualization of the Pex14$_{NTD}$ binding site in relation to the TPR domain of Pex5. The structure revealed that the Wxxx(F/Y) motif forming the binding site for Pex14$_{NTD}$ is situated within an α-helix which protrudes from the TPR domain, which helix has not been defined in previous studies[8,22]. Such docking of Pex14$_{NTD}$ at Wxxx(F/Y) brings Pex14$_{NTD}$ in close proximity to the TPR4 motif of Pex5 and modulates the affinity of Pex5 for PTS1. The Pex5-Pex14$_{NTD}$ interface, however, lacks canonical features of protein-protein interactions[34] and rather constitutes a proximity assembly. In support, residues involved in the assembly contribute relatively weakly to affinity, as demonstrated by analyzing relevant mutants.

There has been ongoing discussion on the role of Pex14 in cargo release[39]. Earlier studies proposed that the binding of Pex14$_{NTD}$ to the Pex5-cargo complex triggers cargo release[23]. In contrast, it has been shown in *Arabidopsis* that the binding of Pex14$_{NTD}$ to Pex5 is not

sufficient for the cargo release[40]. Our complex clearly demonstrates that Pex14$_{NTD}$ binding at the proximal Wxxx(F/Y) site of Pex5 is not sufficient to release MDH cargo. MDH and Pex14$_{NTD}$ bind to Pex5 at nonoverlapping sites, and Pex14$_{NTD}$ binding does not significantly influence the structure of the Pex5 TPR domain, which explains why Pex14$_{NTD}$ cannot induce cargo release by binding to the proximal Wxxx(F/Y) motif. Affinity data (FP) even show a moderate increase in Pex5 affinity for PTS1 in the presence of Pex14$_{NTD}$. We speculate that it is because Pex14$_{NTD}$ binding locks the TPR domain in cargo-bound conformation[12,32]. Additionally, recent literature suggests another mechanism where cargo release is achieved by Pex5 unfolding during recycling through Pex2/10/12 ubiquitin ligase complex[41], while the binding of Pex14$_{NTD}$ allows extraction of the cargo complex from the translocation pore[15].

N-terminal domain (NTD) of Pex5 remains intrinsically disordered. Prior low-resolution SAXS study suggested that Pex5 NTD does not undergo significant rigidification upon saturation with Pex14$_{NTD}$, but rather remains in a disordered state[42]. Our study supports the earlier conclusions—we have not seen any density accounting for Pex5 NTD in any of the maps, though full-length Pex5 was used for complex reconstitution, and the amount of Pex14$_{NTD}$ used for reconstitution was equimolar relative to Wxxx(F/Y) sites. These results indicate that at the tested conditions, Pex5 NTD remains for the major part intrinsically disordered. Our structural observation is interesting in the light of a cryo-EM structure of full-length Pex14, where a large rod-like coiled coil domain of Pex14 homotrimer is well resolved[43], while the Pex14$_{NTD}$ domains located on the other side of the peroxisomal membrane remain unresolved. Pex14$_{NTD}$s are positioned in close proximity in the Pex14 homotrimer, which should facilitate binding to the multiple Wxxx(F/Y) motifs in Pex5 NTD, thereby increasing avidity.

Our structural data demonstrates that the binding of single Pex14$_{NTD}$ to the TPR proximal Wxxx(F/Y) motif does not induce structural reorganization of the TPR domain of Pex5. FP data suggest significant reorganization of Pex5 TPR is neither induced when multiple Wxxx(F/Y) sites are occupied by Pex14$_{NTD}$. The question remains, however, if the binding of multiple Pex14$_{NTD}$s in vivo, in the context of the Pex14 trimer, has a different effect on the TPR domain of Pex5. Increasing the complexity, recent evidence suggests interactions of TPR distal Wxxx(F/Y) motifs with Pex13 SH3 domain[44] and with Pex13 YG-motif domain. It is plausible that these TPR distal Wxxx(F/Y) motifs have evolved to preferentially interact with Pex13 rather than Pex14.

In summary, our findings revealed the structural heterogeneity associated with Pex5 binding at MDH and identified secondary interfaces complementing the PTS1-mediated cargo-Pex5 interaction and Wxxx(F/Y)-mediated interface between Pex5-Pex14$_{NTD}$. Mutants within identified secondary interfaces demonstrate their low contribution to affinity, and we speculate that the secondary interfaces rather constitute proximity assemblies that limit the conformational heterogeneity of the interaction to finetune peroxisomal import.

## Methods

### Protein expression and purification

Genes encoding *Trypanosoma cruzi* MDH, full-length Pex5 (residues 1–666, GenBank PBJ69826.1), truncated Pex5 containing an extended TPR domain containing a single Wxxx(F/Y) motif (residues 314–666; Pex5$_{eTPR}$), and Pex14$_{NTD}$ (residues 21–85) were synthesized de novo with N-terminal $6 \times$ His, $6 \times$ His and $6 \times$ His-TEV site tag, respectively. Genes were cloned in pET24(+) vector using NdeI and XhoI restriction sites. Recombinant vectors were transformed individually into *E. coli* BL21 (DE3) for overexpression. Proteins were purified by Ni-affinity (Ni-NTA) and size exclusion chromatography (SEC) (Supplementary Methods). Purified Pex14$_{NTD}$ was subjected to TEV protease treatment to remove the $6 \times$ His-tag. Mutations were introduced by site-directed mutagenesis, and the mutants were purified as wild-type proteins.

Details of all the protein constructs used in this study are provided in Supplementary Table 4.

### In vitro reconstitution and purification of the ternary complex

Purified MDH (tetramer), full-length Pex5 and Pex14$_{NTD}$ were mixed at 1:4:12 molar stoichiometry and incubated overnight at 4 °C for ternary complex formation. The formed complex was separated from the remaining individual components by size exclusion chromatography (SEC) using Superose6 10/300.

### Analysis of the complex formation

The highest molecular weight peak after the void volume of Superose6 10/300 was collected as the potential ternary complex sample and analyzed by SDS-PAGE. 10 μL of the sample was mixed with 2x SDS-PAGE sample loading buffer and run on the 15% SDS-PAGE gel along with the molecular weight marker in Tris-Glycine running buffer, followed by the Coomassie blue staining. The presence of all components was confirmed from the resultant band pattern (Supplementary Fig. 1).

### Cryo-EM of MDH · Pex5 · Pex14$_{NTD}$ ternary complex

Cryo-EM data collection: Purified ternary complex (~0.75 mg/mL) was vitrified on holey-carbon grids (Quantifoil R 2/1) using FEI Vitrobot Mark IV. Grids were glow-discharged at 8 mA for 70 s using the LEICA EM ACE200 glow-discharger. Glow-discharged grids were loaded with 3 μL of sample, blotted (blot time: 5 s; blot force: 5 units) and flash-frozen by plunging into liquid ethane. Frozen grids were screened for the quality of ice and particle density on 200 kV Glacios microscope equipped with Falcon 4 detector prior to data collection on 300 kV Titan Krios G3i Cryo-TEM microscope equipped with Falcon 4 detector at a magnification of ×105k. Each exposure was recorded into 40 dose-fractioned frames with a total electron dose of 42.25 e⁻/Å² with a pixel size of 0.86 Å/px. Screening and data collection were performed at the cryo-EM facility at the Solaris synchrotron in Krakow, Poland.

Cryo-EM data processing: Data were processed using cryoSPARC[45]. Original pixel size 0.86 Å/px was used throughout the processing. Micrographs were motion corrected and subjected to contrast transfer function (CTF) estimation using patch motion correction and patch CTF estimation jobs, respectively. Protein particles were picked on a subset of micrographs using blob-picker. Picked particles were subjected to template-free 2D classification to generate initial 2D templates. Generated 2D templates were used to pick particles from the whole dataset by template-picker with parameters: particle diameter, 150 Å; Lowpass filter to apply (template/micrograph), 20 Å/20 Å; Angular sampling, 5⁰; Minimum separation distance, 75 Å; Maximum number of picks to consider, 2000. Bad particles were removed by iterative rounds of 2D classification. We first followed the data processing procedure not accounting for conformational variability. Particles in selected classes representing MP1, MP2, and MP3/4 complex were used for initial model generation using ab-initio 3D reconstruction and subsequent refinement using homogenous and heterogeneous refinement jobs in cryoSPARC. This resulted in the maps for MP1 and MP2 showing satisfactory density for MDH, but blurred density for the Pex region. The 3D-variability analysis (3DVA)[30] was used to classify the particles based on heterogeneity. Selected particle clusters and associated volume maps were individually subjected to non-uniform homogenous refinement in order to reconstitute individual 3D maps representing different compositional and conformational states of the complex, i.e., MP1-c, MP1-d, MP2-c, and MP2-d (see Results for the definition of states). Local refinement using the local mask on the Pex region was performed to improve the map within the Pex region. Local mask was generated to include the density of a single copy of the Pex region in both distal and close states of MP1-c, MP1-d, MP2-c, and MP2-d complexes. All particles, including particles in the class representing MP3/4, were used for the local refinement

using this mask, resulting in well-resolved density for Pex14$_{NTD}$. Locally refined Pex region map was combined with the consensus map using Combine map tool in Phenix to generate the final map of MP1P complex[46]. Density maps were further sharpened by B-factor-based sharpening in cryoSPARC and using EMReady[47]. Overall quality of cryo-EM data and data-processing workflow is depicted in Supplementary Figs. 2–6, 14 and Supplementary Table 3.

Model building and refinement: Crystal structure of *T. cruzi* MDH (PDB id: 7QOZ) and Pex5 TPR domain (PDB id: 8OS1) were used to fit in MP1 and MP2 maps using Dock in the map program of Phenix suite[46]. Docked models were subjected to flexible fitting into the map using the namdinator web-server[48]. Additional residues were manually added to the Pex5 TPR model at the N-termini as guided by the density map. The procedure resulted in the final model of Pex5 comprising residues 327–462 and 487–653. Fitted models were refined against the map by iterative cycles of automatic refinement using phenix real-space refinement[49] and interactive refinement using COOT[50] until reasonable geometry and map-to-model correlation were achieved. For the MP1P structure, the refined models of MDH and Pex5 from the above structures were rigid-body fitted in the map and further refined. The Pex14$_{NTD}$ was then de novo built in the map to complete the MDH-Pex5-Pex14$_{NTD}$ model. For Pex14$_{NTD}$ de novo model building, its alphafold[51]-predicted structure was docked within the map. The model was trimmed and refined as needed according to the map by COOT[50], phenix real space refine[46] and ISOLDE[52]. Validations of model and model vs. data for all structures are summarized in Supplementary Figs. 3–6, 14 and Supplementary Table 3.

### Cryo-EM structure determination of MDH-Pex5 binary complex and MDH alone

The MDH-Pex5 binary complex was in vitro reconstituted similarly to the MDH-Pex5-Pex14$_{NTD}$ ternary complex, but without Pex14$_{NTD}$. The cryo-EM structure of MDH alone was solved from a dataset containing MDH-Pex5-Pex14$_{NTD}$ by selection of particles that contained only the MDH tetramer core. The atomic structures were derived as described for the ternary complex. The MDH-Pex5 structure was used to verify the assignment of the density to the Pex14$_{NTD}$ in the MP1P map. The MDH structure was used to verify the assignment of the density to the Pex5$_{457-489}$ loop in the MP1-d and MP2-d maps.

### Data analysis and representation

The analysis, manipulation and representation of cryo-EM maps and 3D structures were performed using COOT[50], PyMOL (Schrödinger, LLC), UCSF Chimera[53] and chimeraX[54]. The twist and tilt angles describing the movement of Pex5 relative to MDH were estimated for the close model with reference to the distal model, while both models were aligned on the MDH structure. The twist angle was determined by the measure rotation tool in chimeraX[54], which reports the rotation between the coordinate systems of the two models with respect to the imaginary axis shown in Supplementary movie 8. We define the tilt angle as the angle between the imaginary 2D planes encompassing selected atoms from Wxxx(F/Y) containing helix and α15 helix of Pex5 in distal and close states[54]. Sequence alignments were performed and presented using MEGA X software and ESpript server[55], respectively.

The statistical significance of differences in compared values was evaluated using Welch's t-test (two-tailed). The *p* value of below 0.05 was considered significant.

### Isothermal titration calorimetry (ITC)

ITC measurements were performed in 20 mM sodium phosphate buffer containing 100 mM NaCl at pH 6.5 (Pex5$_{eTPR}$–Pex14$_{NTD}$ interaction) or 100 mM Hepes buffer containing 150 mM NaCl and 2 mM β-mercaptoethanol at pH 7.5 (MDH-Pex5$_{eTPR}$ interaction) using a Nano ITC 2 G (TA Instruments). Before the experiments, proteins were extensively dialyzed against the relevant buffer. Where relevant, the complex was prepared post-dialysis by mixing Pex5$_{eTPR}$ (20 μM) with the PTS1 peptide (NH$_2$-ARSKL-COOH; 50 μM) and the mixture was used for ITC. The binding experiment involved 5 μL injections of Pex14$_{NTD}$ (175 μM) or Pex5$_{eTPR}$ solution (250 μM) into the calorimeter cell containing Pex5$_{eTPR}$ + PTS1 (20 μM) or MDH (25 μM), respectively. Injections were performed at 300 s intervals. All experiments were conducted at 25 °C with a stirring rate of 250 rpm. The heat of dilution was determined by titration of Pex14$_{NTD}$ or Pex5$_{eTPR}$ to protein-free buffer. The data were analyzed using NanoAnalyze software (TA instruments). The $N$ value (defined as $N = St*(AF_{cell}/AF_{syr})$, where St reflects stoichiometry [number of binding sites], and AF represents active fractions in cell and syringe), the dissociation constant ($K_d$) and the enthalpy change (ΔH) were obtained from fitting the data with a binding model. The Gibbs free energy (ΔG) and change in entropy (ΔS) were calculated using $-RT \ln K_d = \Delta G = \Delta H - T\Delta S$, where $R$ is the gas constant and $T$ is the absolute temperature.

### Fluorescence polarization assay

Fluorescence polarization (FP) was determined using Tecan Infinite Pro F200 plate reader with the 485 nm excitation and 535 nm emission filters. The fluorescence intensities parallel and perpendicular to the plane of excitation were determined in Corning black 96-well NBS assay plates at room temperature. Fluorescence polarization values were expressed in millipolarization units (mP).

In FP binding experiments, Pex5$_{eTPR}$ (9.5 μM) and Pex5 (2.8 μM) were serially diluted in the presence and absence of Pex14$_{NTD}$ (34 μM, constant concentration). 6-FAM labeled PTS1 peptide was added at 10 nM concentration. The FP values were determined and plotted against the log$_{10}$ of Pex5 concentration, and the dissociation constant ($K_d$) was obtained by fitting the experimental data using the following equation:

$$FP = FP_{min} + ((FP_{max} - FP_{min}) * c)/(K_d + c) \qquad (1)$$

where, FP is the experimentally determined fluorescence polarization, FP$_{max}$ represents maximum value of fluorescence polarization (PTS1 peptide saturated with Pex5/Pex5$_{eTPR}$), FP$_{min}$ represents a baseline level (PTS1 peptide alone), $K_d$ is the dissociation constant, and $c$ is the protein concentration. All experiments were repeated three times, $K_d$s were determined independently for each repeat and averaged.

### Reporting summary

Further information on research design is available in the Nature Portfolio Reporting Summary linked to this article.

## Data availability

The cryo-EM maps, generated in this study, have been deposited in the Electron Microscopy Data Bank (EMDB) under accession codes EMD-40056 (MP1P complex), EMD-40003 (MP1-c complex), EMD-40008 (MP1-d complex), EMD-40031 (MP2-c complex), EMD-40032 (MP2-d complex), EMD-50340 (MDH-Pex5 complex) and EMD-50339 (MDH alone). The atomic coordinates, generated in this study, have been deposited in the Protein Data Bank (PDB) under accession codes, 8GI0 (MP1P complex), 8GGD (MP1-c complex), 8GGH (MP1-d complex), 8GH2 (MP2-c complex), 8GH3 (MP2-d complex), 9FEF (MDH-Pex5 complex), and 9FEE (MDH alone). The structures of individual proteins used in this study are available at the Protein Data Bank (PDB) under accession codes, 7QOZ (MDH) and 8OS1 (Pex5 TPR). Source data are provided with this paper.

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

## Acknowledgements

This research was funded by National Science Centre, Poland, in parts by grants 2017/26/M/NZ1/00797 (construct preparation, crystal structures of selected components) and 2020/39/B/NZ1/01551 (remaining experiments), both to GD. R.R.S. was a recipient of NAWA Ulam scholarship. We thank Marcin Jaciuk, Katarzyna Pustelny and Rahul Mehta from Malopolska Centre of Biotechnology of the Jagiellonian University, and Grzegorz Popowicz, Valeria Napolitano and Florian Schlauderer from Helmholtz Center in Munich for valuable discussion, assistance in experiments and providing materials. We acknowledge the access to cryogenic electron microscope at National Synchrotron Radiation Centre SOLARIS supported under contract nr 1/SOL/2021/2 from Ministry of Science and Higher Education, Poland. We thank Michal Rawski and Paulina Indyka for assistance. We acknowledge Polish high-performance computing infrastructure PLGrid (HPC Center: ACK Cyfronet AGH) for providing computer facilities and support within computational grants no. PLG/2022/015912, PLG/2024/017665 and PLG/2025/018725. We thank Klemens Noga from AGH Cyfronet for excellent assistance. We thank Edward H. Egelman (University of Virginia) for making computational facility available for cryo-EM data processing. We thank the MCB Structural Biology Core Facility (supported by the TEAM TECH CORE FACILITY/2017-4/6 grant from Foundation for Polish Science) for providing instruments and support. We acknowledge the Core Facility for Crystallography and Biophysics, University of Warsaw (supported by the Foundation for Polish Science under the European Regional Development Fund, TEAM TECH Core Facility POIR.04.04.00-00-31DF/17) for valuable support.

## Author contributions

R.R.S. devised expression and purification protocols, reconstituted the complex, solved the cryo-EM structure of the ternary complex, analysed the data, wrote the manuscript, and prepared the figures. O.L. solved the cryo-EM structures of MDH-Pex5 and MDH alone. A.B. expressed proteins, performed biochemical assays, analysed the results, and contributed to writing parts of the manuscript. M.J.-R. performed isothermal titration calorimetry (ITC) and analysed the data. O.L., S.P., and T.S. expressed and purified proteins and performed biochemical assays. G.D. obtained funding, designed and coordinated the project, discussed the results, and wrote the manuscript. All authors analyzed the results and approved the final manuscript.

## Competing interests

The authors declare no competing interests.
