## [Transparent Peer Review file · Nature Communications]

Structure of Trypanosoma peroxisomal import complex unveils conformational heterogeneity

Corresponding Author: Professor Grzegorz Dubin

Version 1:

Reviewer comments:

Reviewer #1

(Remarks to the Author)

This review is on a revised version of the manuscript entitled “Structure of Trypanosoma peroxisomal import complex unveils conformational dynamics”, which includes a file on the main manuscript, another one on Supplementary Material, and an extensive rebuttal letter. As it turned out to be impossible to track back various changes mentioned in the rebuttal letter, the reviewer asked for a version of the manuscript with the changes highlighted (it would have been appreciated if such version had been provided without explicit request). As it became evident that large parts of the manuscript were completely rewritten (including the title), the reviewer decided to use this new version as an independent one and to review it from scratch. I will also respond to some comments made in the rebuttal letter.

A major point the authors are aiming for is a claim of distinct swinging motions of the Pex5 receptor upon binding to the cargo used in this contribution (MDH). An alternative interpretation would be to consider a quasi-continuous ensemble of slightly different conformational states, within the limits found in the total of 20 particle clusters (Supplementary Figure 2). The authors suggest the second scenario in their own words by stating: “The random distribution of particles among the clusters indicates a continuous spectrum of Pex5 swinging motion relative to MDH with no preferential orientations (Supplementary Figure 2).” Unfortunately, there is no further characterisation of the 20 clusters, e.g. in terms of the tilt/twist angles of Pex5/MDH arrangements. Is there any particular reason why the second and third most populated clusters (16, 20) were not used for structural interpretation? If the scenario of a conformational ensemble model is true (which appears to be more likely in the author’s own words, see above), the proposed swinging model is either an inaccurate description or is not existent, unless experimental evidence for swinging is provided. Morph videos are illustrative but do not prove swinging. Clarification of this is crucial to avoid a misconception of the data presented. Please note that there are other systems in biology with distinct swinging motions, based on experimental evidence (for some examples, see review PMID: 34547238). A similar argument applies to a frequently used statement of “conformational dynamics”. It remains unclear what the authors exactly mean with it: if it is conformational dynamics in a sense of change from state A to state B, there is no experimental evidence for it. If it refers to conformational wobbliness of the CTD of the Pex5 receptor, it is probably closer to the truth, but this is a well-established concept for many years, referring to some of the literature cited by the authors.

The authors make excessive use of statements about “canonical” versus “non-canonical” interfaces, addressing the Pex5/MDH interactions and Pex5/Pex14 interactions. As “non-canonical” is used for the description of different, unrelated interfaces, it would be advantageous to simply refrain from such terminology and define them in a more defined way, i.e. PTS1 / non-PTS1 for Pex5/cargo and Wxxx(FY) / non-Wxxx(FY) for Pex5/Pex14. Simplified versions of this would be fine too.

There is still a lack of clarity on Pex5 constructs used in different experiments. While in the rebuttal letter under point 2 it is stated: “All the binding data reported in the revised version is derived at conditions which make the ITC data interpretation in the context of cryo-EM structure more relevant.” In the HIGHLIGHTS section of the manuscript, however, it is stated “The structure of the peroxisomal import ternary complex (MDH-Pex5TPR-Pex14NTD) is determined using cryo-EM.” I remain confused as to what construct was used for different types of experiments. Therefore, I would like to request to use a non-ambiguous nomenclature throughout the experimental results text, providing clarity which Pex5 constructs were used, such

as Pex5(FL) and Pex5(CTD), for instance. The use of the term Pex5TPR is also suboptimal, as this construct exceeds the well-established TPR array in Pex5. Finally, what was the reason to use different Pex5 constructs? Insight into this needs to be transparent.

Are there binding data for the ternary Pex5/Pex14/MDH complex (point 8 of the rebuttal letter)? This is confirmed in the rebuttal letter. In Table 2 and in the text it is mentioned that a "short peptide encompassing PTS1 of MDH (residues 319-ARSKL-323)" was used. Why was this peptide, lacking any secondary binding sites, used instead of MDH? I also could not find any description in the Methods section how this peptide complex was prepared. How was it purified, characterised and what was its concentration, when used in experiments? Is there any difference in Pex14 binding in the presence/absence of the peptide? What would be the difference in the presence/s absence of MDH?

Any statistical significance analysis of the binding data (Tables 1 and 2) is missing. In the absence of this, the interpretation of the observed changes is more a matter of belief than based on statistical evidence. Such analysis needs to be added to allow their quantitative interpretation. This is essential as quite a number of binding data are either unexpected or inconclusive. In general, these data need to be better explained. In the absence of estimated statistical significance, they may put the correctness of some of the structural data into question.

Although publishers are consistently request to refrain from superiority claims, both in a sense of superiority of own data and inferiority of others, the manuscript repeats inferiority claims multiple times by classifying previous findings as "poor", directly or indirectly suggesting superiority of their own data (see e.g. reply to point 13 in the rebuttal letter). The data presented do not explain why Pex5-cargo binding facilitates translocation into the peroxisomal lumen (abstract), a "process poorly understood". Therefore, despite very interesting structural insights, I do have doubts that they really facilitate an improved understanding of the peroxisomal cargo translocation process.

Supplementary items are not cited in sequential order. This needs to be corrected.

Additional comments:

Please remove the statement "the causative agent of human Chagas disease" from the abstract. This is repeated three times in the manuscript. The data presented neither link to nor explain why *Trypanosoma cruzi* is the causative agent of human Chagas disease.

Introduction, p. 4: "What is established is that the import of the Pex5-cargo complex involves assistance from membrane-spanning receptors, Pex14 and Pex13(5)." Neither Pex14 nor Pex13 are described as "receptors" in reference 5 nor, to the best of my knowledge, in other relevant literature. Pex5 is the receptor.

Results, section "Cryo-EM reconstructions and heterogeneity of the ternary complex", page 7: "It is of note that although we observed particles with two Pex5 molecules bound on the same side of the MDH tetramer, we did not observe particles with two Pex5 molecules bound to diagonally opposite MDH subunits (Figure 2B)." Without further explanation, this sentence does not make sense. The inherent MDH symmetry needs to be explained or cited somewhere. Assuming that it is 222, there is no preferred side of the tetramer. Unless the symmetry is different, there is a requirement for two Pex5 molecules in "diagonal" positions in the MP3 complex, even if one of the two is not interpretable in molecular detail.

Results, section "Cryo-EM reconstructions and heterogeneity of the ternary complex", page 7: "The piggybacking of cargo isoforms within heteromultimeric complexes suggests that saturation of all Pex5 interaction sites is not required for translocation29,30." I could not find any discussion on the lack of requirement for saturated binding of cargos in references 29+30.

Results, section "Pex5-binding on MDH is of dynamic nature", page 7: "This interaction is stabilized by a network of hydrogen bonds involving the PTS1 C-terminal carboxylate group, mainchain donors and acceptors, and lysine sidechain amine, as observed previously in different species7." Could this network please be shown for the structure presented here? Is this network around the PTS1 motif preserved in different binding conformations or is it different?

Results, section "Pex5-binding on MDH is of dynamic nature", page 7: "Similar compaction is also observed for *Trypanosoma brucei* and human Pex56-8,31, suggesting that TPR domain compaction upon PTS1 binding is general across species." To the best of my knowledge, the concept of binding site compaction was introduced by Fodor et al., (2015, PMID: 25369882), and confirmed in reference 31, from some of the authors of this paper. Unfortunately, previous work was also not be adequately cited in reference 31.

Results, section "Secondary MDH-Pex5 interfaces does not significantly contribute to affinity", page 9: "Because close analysis of interfaces 1 and 2 does not reveal classical features of protein-protein interaction and because identified swinging motion affects the extent of the interfaces, we wanted to test if the identified interfaces contribute to affinity." What are "classical features of protein-protein interaction"? Please define.

Figure 1:

What is the "standard data processing pipeline"? Please reference.

A. Assuming 222 MDH symmetry (further comments elsewhere), how could it be excluded that a 4th Pex5 component is simply hidden behind MDH because of symmetry?

B. Pex5 and MDH are clearly separated by different colors. There is no need of further encircling.
C, What is the message of this panel? I don't get it.

Figure 2:

A. How was the angular movement (tilt) measured? This needs to be described in the methods section.
B. Is this a movement (translation) or rotation, or a combination of both? Could the movements please be plotted against the visible Pex5 sequence, in an additional supplement figure? Given that the color code of the second Pex5 molecule in MP2 is limited to green/yellow, are the authors suggesting that the flexibility of Pex5 binding of the second Pex5 molecules in MP2 is more restricted than for the first Pex5 molecule?

Figure 3:

A. The mixture of cartoon (MDH) and surface (Pex5) is more confusing than clarifying. Could any specific interactions (hydrogen bonds, salt bridges) be identified, and if so, please indicated? To what extent are the interfaces shown in the different binding conformations (close, distal, intermediate, as explained in S Figure 8) are preserved or different?
B. I don't get the sentence "The density is not seen in a groove distal from Pex5 occupied site (red)." Isn't this logical, if no Pex5 is bound?

Figure 4 (and corresponding text)

It is surprising that mutagenesis of residues, based on the structural model of the interaction by Pex5 with Pex14, shows only little or no effect (for statistical assessment, see comments elsewhere): how sure are the authors about a correct sequence register of Pex14? Is this model supported by AF3 predictions and previous structural data of the Pex14-NTD with a Pex5 peptide. Please elaborate on this, possibly by adding figures in the supplement.

Supplementary Figure 1A: Please characterise all three major peaks of the ternary complex by SDS-PAGE. This is important for clarification of incomplete occupancy of Pex5 in MDH/Pex5 complexes.

I will not comment on all replies of the rebuttal letter here, as the review would become too excessive.

Reviewer #2

(Remarks to the Author)

With the better resolution structure of the MDH:PEX5:PEX14 NTD complex and additional ITC data, the authors have significantly improved the quality of the presented data; however, I still have reservations about whether the data presented advances our understanding of peroxisome import. The importance of the dynamics at the PEX5:MDH interface is limited by the lack of a functional import assay to test these hypotheses. The authors' claim that binding of PEX14 to the proximal WXXXF motif is insufficient to cause cargo release could be an interesting result for the understanding peroxisome import, but the authors do not have quantitative data, for example of PEX5 and PTS1 peptide affinity with and without PEX14 to conclusively make their claim. In summary, I agree that the author's data now support their claims (1 - Cryo-EM structure of Pex5:Pex14 NTD:MDH, 2 - Pex5 – MDH interaction is dynamic, and 3-Pex5-Pex14 NTD site extends beyond canonical site), but I am not convinced these claims advance our understanding of peroxisome import.

Several suggestions that I think the authors could address with changes to the text:

"Furthermore, our results show that the Pex14NTD binding at Pex5 proximal Wxxx(F/Y) site is not enough for structural reorganization of the Pex5 NTD and cargo release in vitro." The reader is not given enough context in the introduction to understand the importance of this statement. It would be helpful to talk about domain architecture of Pex14, topology at the membrane, the need for release of cargo in the peroxisome in the intro. This is well covered in the discussion, but could be discussed more in the introduction.

"The density is present only at sites occupied by Pex5 and not empty PTS1 sites indicating it corresponds to a fragment of Pex5457-489 loop (Figure 3B; Supplementary Figure 12)." It seems possible this density could be a loop of MDH stabilized by the presence of Pex5 instead of a loop of Pex5 stabilized by MDH. Can the authors more clearly state why they think this density can definitively be assigned to PEX5 (ie rule out missing loops in MDH)?

The authors have chosen to exclude their previous ITC data of PEX5 and PEX14 interactions without PTS1 peptide present. This data could be useful for interpreting how PTS1 binding alters PEX5 affinity for PEX14, which is relevant to the authors discussion of PEX14 causing PEX5 to release its cargo.

The authors could discuss the conservation of the non-canonical interfaces for PEX14 NTD and PEX5.

"Proteins were purified by Ni-affinity (Ni-NTA) and size exclusion chromatography (SEC)." Authors should describe buffers, lysis, concentration methods for purification and EM grid preparation.

I have minor comments on typos/phrasing below, if it helps the authors.

Minor comments:

"Pex14 was additionally implemented in cargo extraction from the translocation pore." Phrasing is off - did you mean implicated instead of implemented?

"The maps accounting for the distal states of MDH-Pex5 complex (MP1-d and MP2-d) show additional density proceeding Pex5 residue Ser491 and facing MDH62-70 loop (Figure 3B; Supplementary Figure 12)." Did you mean preceding instead of proceeding?

"To further test the hypothesis, we mutated Pro490 residue" – typo in further

"The density is not seen in a groove distal from Pex5 occupied site (red)." – typo in form

"Prior studies involving ATG and SCP2" – typo in AGT

"again demonstrating that the interactions at the intersubunit groove of MDH unfavorably affect the complex dynamics and affinity." – typos in interactions and intersubunit

Version 2:

Reviewer comments:

Reviewer #1

(Remarks to the Author)

The changes in response to previous referee comments in the current revised version (NCOMMS-23-36304B) are appreciated. Overall, the manuscript has been improved. I am still not really convinced by the concept of swinging motions, as swinging implies directivity, for which there is no evidence. At this point, I think this needs to be left for future discussion, when this paper is published.

Going through the previous points made by this referee, I still request the following changes:

Statistical significance tests are now been mentioned in the text but are still not systematically applied to estimate the significance of the quantitative measurements in Tables 1 and 2. Mentioning the statistical significance in the text narrative is fine but cannot replace a systematic analysis where quantitative data are shown, i.e. Tables 1 and 2.

This is of particular importance, as surprisingly a Pex5 version with a GSGS insert shows an approximately threefold improvement in binding to MDH. The statement "The change in affinity was entropy driven indicating that the secondary interactions have overall negative effect on affinity by restricting the conformational dynamics of the wild type interface", implying that it relieves the restriction on the conformational dynamics of wt Pex5 remains unproven by any complementary experiments. The next sentence in the text "the affinity of MDH(GSGS) is comparable to that of a short peptide encompassing PTS1 of MDH (residues 319-ARSKL-323; $p > 0.1$) (Table 1) indicating that the introduction of the GSGS linker has indeed abolished all secondary interactions at the Pex5-MDH interface" even implies that the presence of the second site in the wild-type context has a negative effect on overall binding. So what is the functional impact of this second site? This remains an open question.

Please remove the term "noncanonical" from the abstract, as the term will not be defined later on in the manuscript.

Unfortunately, the revised nomenclature for Pex5 constructs is still quite confusing. I could not find a definition for the acronym Pex5eTPR. Does the "e" stand for "extended"? According to Supplementary Table 4 Pex5hTPR "Refers to residues within full length Pex5 defined by density in the ternary MDH-Pex5-Pex14NTD complex structure", so the construct behind it is actually full-length Pex5. So, it's full-length Pex5 that was used for experiments, with limited structural visibility to Pex5hTPR. A statement on p. 10 about Pex5eTPR as "a construct roughly equivalent to Pex5hTPR" adds to further confusion. Please make a clear distinction between expressed and purified constructs, and expressing visibilities in structures.

I have noted that the new version of the manuscript contains various new changes that were not requested (or requested by the other referee, at least to some extent). In going through these changes, I have noticed that some new unclear statements have been introduced. At this point, I believe it would be beyond the role of this referee to comment on them in detail, to avoid an unintended role as advisor rather than referee. However, I feel it is important to flag this up to ensure that the final version to be eventually published is free of major deficiencies, ambiguities and/or errors.

Just one example, to illustrate of the problem: Supplementary Figure 18 contains an acronym KdAV. I could not find an explanation for it. The measured values have a dimension, nM^{-1} I think. However, no dimension is given. These things

needs to be corrected.

Reviewer #2

(Remarks to the Author)

The reviewers have addressed my concerns, and I appreciate their added fluorescence anisotropy experiments to assess how PEX14 NTD alters PEX5:PTS1 affinity.

Some additional comments to improve clarity:

The Pex5hTPR is used in the abstract and intro, but isn't clearly defined until the results section. Perhaps in the other locations hTPR could be replaced by the residue numbers or hTPR should be defined earlier.

Supplementary Figure 7 – coloring the shown sidechains by atom (ie red = oxygen) would help clarify the H-bond network.

Figure 2 legend – typo in “confirmations” which should be conformations.

In Results section:

"Pex5eTPR (residues 314-666), a construct roughly equivalent to Pex5hTPR"

- Please define hTPR by residue numbers here

- eTPR notation doesn't match Supplementary Table 2: Pex5eTPR (residue 316-666)

Authors state:

"To disrupt interface 1, we replaced the MDH62-70 loop with GS-linker. This resulted in a moderate decrease (~2.3-fold) ($p < 0.001$) in affinity towards Pex5eTPR indicating little contribution of the interface to affinity." Please state the K_d of the GS linker mutant here or indicate that the MDH62-70 loop with GS-linker is (Delta62-70) in Table 1. When I look at Table 1, it is not immediately clear if the authors are referring to the mutant GSGS or Delta62-70.

The authors discuss the comparison of the affinity between Pex5 and Pex14 NTD in the presence and absence of the PTS1 peptide in the rebuttal letter ("Affinity of Pex14NTD for Pex5eTPR determined by ITC in the presence of PTS1 peptide ($K_d = 53 \pm 12$) is lower ($p < 0.001$) compared to affinity determined in the absence of PTS1 peptide ($K_d = 35 \pm 5$). The statistical significance was determined using the Welch's t-test.", but it isn't discussed in the manuscript (that I could find). I personally find it interesting that the presence of the PTS1 peptide increases the K_d between Pex5eTPR and Pex14 NTD, and would encourage the authors to include the statement from their rebuttal in the manuscript. It is interesting that N353E and Y358A Pex5 mutants do not have the same change in K_d with the addition of PTS1, suggesting that these mutants might decouple PTS1 binding and PEX14 NTD binding.

Point-by-point responses to the reviewers’ comments (R2)**Reviewer #1**

“This review is on a revised version of the manuscript entitled “Structure of Trypanosoma peroxisomal import complex unveils conformational dynamics””

Major comments:

1. *“A major point the authors are aiming for is a claim of distinct swinging motions of the Pex5 receptor upon binding to the cargo used in this contribution (MDH). An alternative interpretation would be to consider a quasi-continuous ensemble of slightly different conformational states, within the limits found in the total of 20 particle clusters (Supplementary Figure 2). The authors suggest the second scenario in their own words by stating: “The random distribution of particles among the clusters indicates a continuous spectrum of Pex5 swinging motion relative to MDH with no preferential orientations (Supplementary Figure 2).” Unfortunately, there is no further characterisation of the 20 clusters, e.g. in terms of the tilt/twist angles of Pex5/MDH arrangements. Is there any particular reason why the second and third most populated clusters (16, 20) were not used for structural interpretation? If the scenario of a conformational ensemble model is true (which appears to be more likely in the author’s own words, see above), the proposed swinging model is either an inaccurate description or is not existent, unless experimental evidence for swinging is provided. Morph videos are illustrative but do not prove swinging. Clarification of this is crucial to avoid a misconception of the data presented. Please note that there are other systems in biology with distinct swinging motions, based on experimental evidence (for some examples, see review PMID: 34547238). A similar argument applies to a frequently used statement of “conformational dynamics”. It remains unclear what the authors exactly mean with it: if it is conformational dynamics in a sense of change from state A to state B, there is no experimental evidence for it. If it refers to conformational wobbliness of the CTD of the Pex5 receptor, it is probably closer to the truth, but this is a well-established concept for many years, referring to some of the literature cited by the authors.”*

Our data documents the conformational heterogeneity of Pex5_{hTPR} (abbreviation refers to the resolved part of Pex5 in cryo-EM map) orientations relative to MDH cargo in the structure of MDH/Pex5_{hTPR} /Pex14_{NTD} complex. In further top-down analysis the heterogeneity could be interpreted as snapshots of a continuous motion between boundary conditions or a “*quasi-continuous ensemble of slightly different conformational states*”, but our data does not allow to distinguish between the two models. What our 3DVA results show, is the presence of particles across the whole conformation space between the close and distal states of the complex (Supplementary Figure 2). This is consistent with a relatively low energy contribution of PTS1-independent interactions. However, the number of clusters resulting from 3DVA analysis is shaped by input parameters and should not be interpreted as a number of discrete states. Our data does not distinguish between continuous, quasi-continuous or discrete distribution of Pex5_{hTPR} orientations relative to MDH. The best model explaining the data assumes ensemble of states that are in dynamic equilibrium with low-energy barrier between the states and we refer to this model as “conformational dynamics”. This model does not conclude on the lifetime or relative population of states. We now clearly introduce the term and discuss the above considerations in the revised manuscript (R2).

For detailed characterization we chose the boundary conditions (the most “distal” and most “close” conformations) and not “*second and third most populated clusters (16, 20)*”. The choice was deliberate and made to delineate Pex5_{hTPR}/MDH interactions which limit further

reorientation of Pex5_{hTPR} relative to MDH. The populations of the clusters given in Supplementary Figure 2 are based on a single experiment (3DVA of a single cryo-EM dataset collected from a single grid). The figure is presented to illustrate the distribution in this single experiment only, but because the data lacks estimation of interexperiment variation it should not be used to conclude on quantitative population differences (if any) between clusters. Results shown in Supplementary Figure 2 only allow to conclude that particles are present across the whole range of the conformational space between close and distal states.

The morph video shows continuous morphing through all identified clusters and not just between boundary conditions. We agree with the reviewer that this does not “prove swinging”, but we do not make such claim. The video demonstrates that no significant outliers are present among clusters. Because clusters provide continuous / quasi-continuous representation of the intermediates between boundary states and because clusters are closely interspersed, the detailed analysis of intermediate clusters does not add significant information and only the boundary conditions were analyzed.

2. *“The authors make excessive use of statements about “canonical” versus “non-canonical” interfaces, addressing the Pex5/MDH interactions and Pex5/Pex14 interactions. As “non-canonical” is used for the description of different, unrelated interfaces, it would be advantageous to simply refrain from such terminology and define them in a more defined way, i.e. PTS1 / non-PTS1 for Pex5/cargo and Wxxx(F/Y) / non-Wxxx(F/Y) for Pex5/Pex14. Simplified versions of this would be fine too.”*

The suggested changes in terminology were introduced in the revised version according to the reviewer’s suggestion to clearly define the described interfaces. “Noncanonical” is now only used in the abstract and at the conclusion of the discussion as a concise general term to collectively distinguish previously described interactions from those described in this study. The term is not used in the manuscript text with reference to particular Pex5_{hTPR}/MDH or Pex5_{hTPR}/Pex14 interactions.

3. *“There is still a lack of clarity on Pex5 constructs used in different experiments. While in the rebuttal letter under point 2 it is stated: “All the binding data reported in the revised version is derived at conditions which make the ITC data interpretation in the context of cryo-EM structure more relevant.” In the HIGHLIGHTS section of the manuscript, however, it is stated “The structure of the peroxisomal import ternary complex (MDH-Pex5TPR-Pex14NTD) is determined using cryo-EM.” I remain confused as to what construct was used for different types of experiments. Therefore, I would like to request to use a non-ambiguous nomenclature throughout the experimental results text, providing clarity which Pex5 constructs were used, such as Pex5(FL) and Pex5(CTD), for instance. The use of the term Pex5TPR is also suboptimal, as this construct exceeds the well-established TPR array in Pex5. Finally, what was the reason to use different Pex5 constructs? Insight into this needs to be transparent.”*

A non-ambiguous nomenclature of constructs was introduced and consistently used in the revised version of the manuscript (R2) including the highlights. In addition, supplementary table 4 is introduced to summarize the nomenclature. Suboptimal overlap with terminology used in previous literature for the description of slightly different constructs was avoided. The reasons for using truncated Pex5 construct is stated in R2 version.

4. *“(a) Are there binding data for the ternary Pex5/Pex14/MDH complex (point 8 of the rebuttal letter)? This is confirmed in the rebuttal letter. (b) In Table 2 and in the text it is mentioned that a “short peptide encompassing PTS1 of MDH (residues 319-ARSKL-323)” was used. Why was this peptide, lacking any secondary binding sites, used instead of MDH? I also could not find any description in the Methods section how this peptide complex was prepared. How was it*

purified, characterised and what was its concentration, when used in experiments? (c) Is there any difference in Pex14 binding in the presence/absence of the peptide? What would be the difference in the presence/s absence of MDH?"

(a) We provide binding data for Pex5_{eTPR} (residues 314-666, equivalent to Pex5_{hTPR}; contains the proximal WxxxF/Y motif and TPR domain) and wild type MDH (Table 1). We also provide binding data for Pex5_{eTPR} and Pex14_{NTD} in the presence of PTS1 peptide (Table 2). The rebuttal letter confirms that the latter binding affinity is determined in the presence of PTS1. Such experimental setup was requested by Reviewer #2 in comment #2 of the prior round of reviews (R1) to ensure the closed conformation of TPR domain of Pex5. It was not our intention to claim anything other than the above in the prior rebuttal or here.

(b) The PTS1 peptide was chosen according to the suggestion of Reviewer #2 from comment #2 of the prior round of reviews. The particular NH₂-ARSKL-COOH peptide was chosen because of satisfactory solubility according to our prior experience. The complex was characterized by ITC (Table 1). The concentration of the complex used in experiments was stated in materials and methods in section pertaining to ITC. In the revised version of the manuscript we have modified the earlier description of ITC experiment to explicitly state how was the Pex5-PTS1 complex created.

(c) Affinity of Pex14_{NTD} for Pex5_{eTPR} determined by ITC in the presence of PTS1 peptide ($K_d=53\pm 12$) is lower ($p<0.001$) compared to affinity determined in the absence of PTS1 peptide ($K_d=35\pm 5$). The statistical significance was determined using the Welch's t-test. The tetrameric form of MDH significantly complicates the model and we have not attempted to determine the affinity of Pex14_{NTD} for (Pex5_{eTPR})₄(MDH tetramer). We however demonstrated that interaction with Pex14_{NTD} modulates the PTS1 affinity of Pex5_{eTPR} ($K_d=254\pm 9$ nM and $K_d=80\pm 5$ nM; $p < 0.001$) in the absence and presence of Pex14_{NTD}, respectively) and Pex5 ($K_d=88\pm 5$ nM and $K_d=37\pm 2$; $p < 0.001$), consistent with direct interaction of Pex14_{NTD} with TPR domain of Pex5. This data is included and discussed in the revised version of the manuscript.

5. *"Any statistical significance analysis of the binding data (Tables 1 and 2) is missing. In the absence of this, the interpretation of the observed changes is more a matter of belief than based on statistical evidence. Such analysis needs to be added to allow their quantitative interpretation. This is essential as quite a number of binding data are either unexpected or inconclusive. In general, these data need to be better explained. In the absence of estimated statistical significance, they may put the correctness of some of the structural data into question."*

Analysis of statistical significance of differences in binding data among tested constructs has been introduced in the revised version of the manuscript. p-values have been reported to support the described quantitative interpretations.

6. *"Although publishers are consistently request to refrain from superiority claims, both in a sense of superiority of own data and inferiority of others, the manuscript repeats inferiority claims multiple times by classifying previous findings as "poor", directly or indirectly suggesting superiority of their own data (see e.g. reply to point 13 in the rebuttal letter). The data presented do not explain why Pex5-cargo binding facilitates translocation into the peroxisomal lumen (abstract), a "process poorly understood". Therefore, despite very interesting structural insights, I do have doubts that they really facilitate an improved understanding of the peroxisomal cargo translocation process."*

We did not intend to claim superiority/inferiority of any our/prior data. What we intended was to state that some of the processes still require further clarification which would substantiate the results of prior research while building on those results. The issues referred to by the reviewer are our unintentional linguistic errors which we have removed from the revised version. Alongside we have toned down the too broadly defined claims referred to by the reviewer.

7. *“Supplementary items are not cited in sequential order. This needs to be corrected.”*

The supplementary items are now cited in sequential order in the revised version of the manuscript (R2).

Additional comments:

a. *“Please remove the statement “the causative agent of human Chagas disease” from the abstract. This is repeated three times in the manuscript. The data presented neither link to nor explain why Trypanosoma cruzi is the causative agent of human Chagas disease.”*

The statement was removed from the revised abstract.

b. *“Introduction, p. 4: “What is established is that the import of the Pex5-cargo complex involves assistance from membrane-spanning receptors, Pex14 and Pex13 (5).” Neither Pex14 nor Pex13 are described as “receptors” in reference 5 nor, to the best of my knowledge, in other relevant literature. Pex5 is the receptor.”*

Reference to ‘receptors’ was replaced by ‘proteins’ in the revised manuscript.

c. *“Results, section “Cryo-EM reconstructions and heterogeneity of the ternary complex”, page 7: “It is of note that although we observed particles with two Pex5 molecules bound on the same side of the MDH tetramer, we did not observe particles with two Pex5 molecules bound to diagonally opposite MDH subunits (Figure 2B).” Without further explanation, this sentence does not make sense. The inherent MDH symmetry needs to be explained or cited somewhere. Assuming that it is 222, there is no preferred side of the tetramer. Unless the symmetry is different, there is a requirement for two Pex5 molecules in “diagonal” positions in the MP3 complex, even if one of the two is not interpretable in molecular detail.”*

We appreciate the reviewer pointing to our invalid description. We have accordingly modified the text in the revised manuscript (including explicitly citing the symmetry of the complex) and made relevant modifications to Figure 1 to clearly illustrate the related manuscript text.

d. *“Results, section “Cryo-EM reconstructions and heterogeneity of the ternary complex”, page 7: “The piggybacking of cargo isoforms within heteromultimeric complexes suggests that saturation of all Pex5 interaction sites is not required for translocation 29,30.” I could not find any discussion on the lack of requirement for saturated binding of cargos in references 29+30.”*

Our description was unclear at that point. What we meant was that it is uncertain whether binding of more than one Pex5 molecule is necessary for translocation of the oligomeric cargo. The piggybacking of cargo isoforms lacking PTS1 within heteromultimeric complexes^{29,30} suggests that the binding of Pex5 to all subunits of the oligomer is not required for translocation.

The cited literature refers to piggybacking only, while the discussion on the required number of Pex5 molecules is specific for this manuscript.

The description is now modified accordingly in the revised version of the manuscript.

e. *“Results, section “Pex5-binding on MDH is of dynamic nature”, page 7: “This interaction is stabilized by a network of hydrogen bonds involving the PTS1 C-terminal carboxylate group, mainchain donors and acceptors, and lysine sidechain amine, as observed previously in different species7.” Could this network please be shown for the structure presented here? Is this network around the PTS1 motif preserved in different binding conformations or is it different?”*

The referenced hydrogen bond network has been depicted in Supplementary Figure 7 for both the distal and closed states. The rearrangement of this network is evident between the distal and close states and now described in the revised manuscript text with depiction in added Supplementary Figure 7.

f. *“Results, section “Pex5-binding on MDH is of dynamic nature”, page 7: “Similar compaction is also observed for Trypanosoma brucei and human Pex5 6-8,31, suggesting that TPR domain compaction upon PTS1 binding is general across species.” To the best of my knowledge, the concept of binding site compaction was introduced by Fodor et al., (2015, PMID: 25369882), and confirmed in reference 31, from some of the authors of this paper. Unfortunately, previous work was also not be adequately cited in reference 31.”*

The reference Fodor et al. (2015, PMID: 25369882) was accidentally omitted in the referenced fragment of the text (the work was cited elsewhere in this manuscript in mentioned context). The reference is now included in the revised version of the manuscript in the context of the binding site compaction in the place indicated by the reviewer.

g. *“Results, section “Secondary MDH-Pex5 interfaces does not significantly contribute to affinity”, page 9: “Because close analysis of interfaces 1 and 2 does not reveal classical features of protein-protein interaction and because identified swinging motion affects the extent of the interfaces, we wanted to test if the identified interfaces contribute to affinity.” What are “classical features of protein-protein interaction”? Please define.”*

In a simplified view, but still sufficient for the purpose of the discussion, protein-protein interactions are driven largely by hydrophobic effect which requires certain surface complementarity. Often, the more dynamic the interaction, the higher the modulation by buried charged and polar residues. For a detailed discussion the reader is directed in the revised version of the manuscript to a review by Keskin et al. (Chem. Rev. 2008, 108, 1225-1244). The MDH-Pex5_{hTPR} interfaces 1 and 2 show neither significant surface complementarity, hydrophobicity nor charged or polar residues aligned for mutual interactions. Therefore, they do not resemble protein-protein interaction sites, and are rather defined in this work as proximity regions.

h. Figure 1

(i) *“What is the “standard data processing pipeline”? Please reference.”*

The term has been removed in the revised version of the manuscript. We now clearly refer to two different data treatments used, that is “data processing procedure not accounting for conformational variability” and “3D-variability analysis”. The approaches are described in details in the revised Methods section.

- (i) *“Panel A: Assuming 222 MDH symmetry (further comments elsewhere), how could it be excluded that a 4th Pex5 component is simply hidden behind MDH because of symmetry?”*

Thank you for pointing out our inconsistent reasoning. The third and fourth copies of Pex5 can be superimposed in 2D class average and thus cannot be differentiated. We have now named the third subset as MP3/4 in the revised manuscript to reflect the indistinguishable character of the two configurations.

- (ii) *“Panel B: Pex5 and MDH are clearly separated by different colors. There is no need of further encircling.”*

The circles are removed in the revised manuscript.

- (iii) *“Panel C: What is the message of this panel? I don’t get it.”*

The panel shows that all 4 PTS1 sites within MDH tetramer are equally accessible, that is, no clear steric occlusion is present at any of the sites which could preclude Pex5 binding. The figure caption was revised accordingly to clearly point to the message of the panel.

i. Figure 2

- (i) *“How was the angular movement (tilt) measured? This needs to be described in the methods section.”*

Relevant description was included in the revised methods section.

- (ii) *“(a) Is this a movement (translation) or rotation, or a combination of both? (b) Could the movements please be plotted against the visible Pex5 sequence, in an additional supplement figure? (c) Given that the color code of the second Pex5 molecule in MP2 is limited to green/yellow, are the authors suggesting that the flexibility of Pex5 binding of the second Pex5 molecules in MP2 is more restricted than for the first Pex5 molecule?”*

(a) The movement is described in detail in section “Pex5-binding on MDH is of dynamic nature”: “(...) This swinging motion can be characterized by two components: tilting and twisting. Pex5_{hTPR}, tilts nearly as a rigid body, oscillating between close and distal conformations with a maximum tilt angle of 17° relative to the PTS1 (Figure 2A; Supplementary Movies 1-7). Additionally, during this tilting motion, Pex5_{hTPR} twists up to 13° relative to the axis roughly parallel to the TPR3 motif (Supplementary Movie 8). The combined tilting and twisting motions result in a radial displacement of Pex5_{hTPR} ranging between 3 to 8 Å (Figure 2).

(b) The supplementary figure suggested by the reviewer and depicting the residue-by-residue movements plotted against the Pex5 sequence was included as Supplementary figure 8.

(c) The fact that the flexibility of the second Pex5_{hTPR} molecule in MP2 is more restricted compared to the first Pex5_{hTPR} molecule stems directly from the experimental data. The color coding in panels B and C reflects comparison of closed and distal conformations without any assumptions or data manipulation. Closed and distal conformations are overlaid based on MDH and colors represent the difference in position of corresponding C_α atoms between the two states.

j. Figure 3

- (i) *"The mixture of cartoon (MDH) and surface (Pex5) is more confusing than clarifying. Could any specific interactions (hydrogen bonds, salt bridges) be identified, and if so, please indicated? To what extent are the interfaces shown in the different binding conformations (close, distal, intermediate, as explained in S Figure 8) are preserved or different?"*

The figure was modified in line with the reviewer comment. We have created renderings with and without the surface representation and decided that to best incorporate the reviewer request and retain figure clarity a semi-transparent surface model with selected residues in stick model is most relevant to depict all the details discussed in the text. The interface is described in the section "Pex14_{NTD} binding site and noncanonical interactions with Pex5" and the involved residues are now depicted and labeled in the revised version of the figure, but no specific interactions are involved in these proximity assemblies as discussed in the text (see also response to additional comment "g" above). Corresponding figure for proximal conformation is provided in revised Supporting Information (Supplementary figure 12). The interfaces are largely preserved in close and distal conformations.

- (ii) *"I don't get the sentence "The density is not seen in a groove distal form Pex5 occupied site (red)." Isn't this logical, if no Pex5 is bound?"*

The sentence was introduced in response to comment #10 of reviewer #1 in the prior (R1) round of reviews. The reviewer requested supporting evidence that the density is indeed a fragment of Pex5 loop. Apart from additional experimental evidence (MDH structure in the absence of Pex5) the revised text refers to the fact that the density is present only in the presence of Pex5 and not the unoccupied sites. The conclusion is "logical", but is stated there to attract readers attention to the fact which otherwise might went overlooked.

- k. *"Figure 4 (and corresponding text): It is surprising that mutagenesis of residues, based on the structural model of the interaction by Pex5 with Pex14, shows only little or no effect (for statistical assessment, see comments elsewhere): how sure are the authors about a correct sequence register of Pex14? Is this model supported by AF3 predictions and previous structural data of the Pex14-NTD with a Pex5 peptide. Please elaborate on this, possibly by adding figures in the supplement."*

The fit of Pex14_{NTD} into the density is unambiguous in terms of sequence register. This is because the Pex5_{hTPR} is rigid and well defined by density and the Wxxx(F/Y) motif interaction with C-terminal domain of Pex14_{NTD} is well defined by prior structures. These two facts, even in absence of any density accounting for Pex14 would position Pex14 exactly in the place in which it is found in our structure. Our structure, however, has a clear density to account for the helices comprising Pex14_{NTD}. Because the helices are not parallel, but located at various angles to each other, it is impossible to shift the rigid body of Pex14 within the density derived from cryo-EM without violating the fit of some helices. The above, together with availability of previous structures of Pex14_{NTD} makes the sequence register assignment unambiguous.

The resolution in the referenced region does not allow to see particular sidechains, but rigid nature of Pex14_{NTD} and available structures allow to predict with high confidence the direction at which the sidechains point.

AlphaFold3 places the entire Pex5 helix containing the Wxxx(F/Y) motif in position which is not supported by experimental data. The helix is clearly defined by density in our structure and

extends from the TPR domain in a different orientation than suggested by AlphaFold3. Accordingly, Pex14_{NTD} is positioned in a completely different place than seen in experimental data. This requires a conclusion that in this particular case AlphaFold3 fails to predict the correct structure with regard to the orientation of Pex14_{NTD} relative to Pex5.

As suggested by reviewer, we compared the Pex14_{NTD} structure bound to the Pex5 Wxxx(F/Y) containing motif helix (part of MP1P complex structure PDB id: 8gi0 of the present study) with the previously described Pex14-Pex5 structure (PDB id: 2w84). The RMSD between these two structures is minimal (~1.0 Å between equivalent 46 C α atoms) suggesting the significant similarity between the two and thus indicating the correctness of Pex14_{NTD} assignment to the density. A supplementary Figure 15 is added to show this comparison.

I. *“Supplementary Figure 1A: Please characterise all three major peaks of the ternary complex by SDS-PAGE. This is important for clarification of incomplete occupancy of Pex5 in MDH/Pex5 complexes.”*

The requested data is provided in the revised Supplementary Figure 1.

Reviewer #2

Major comments:

1. *“With the better resolution structure of the MDH:PEX5:PEX14 NTD complex and additional ITC data, the authors have significantly improved the quality of the presented data; however, I still have reservations about whether the data presented advances our understanding of peroxisome import. The importance of the dynamics at the PEX5:MDH interface is limited by the lack of a functional import assay to test these hypotheses. The authors’ claim that binding of PEX14 to the proximal WXXXF motif is insufficient to cause cargo release could be an interesting result for the understanding peroxisome import, but the authors do not have quantitative data, for example of PEX5 and PTS1 peptide affinity with and without PEX14 to conclusively make their claim. In summary, I agree that the author’s data now support their claims (1 - Cryo-EM structure of Pex5:Pex14 NTD:MDH, 2 - Pex5 – MDH interaction is dynamic, and 3-Pex5-Pex14 NTD site extends beyond canonical site), but I am not convinced these claims advance our understanding of peroxisome import.”*

The data presented in our work supports the three claims enumerated by the reviewer:

- (1) Structure of MDH:Pex5:Pex14_{NTD} assembly demonstrates how the three components interact together forming the translocation/post-post translocation complex
- (2) The extent of the dynamics of the Pex5-cargo interaction, a characteristic unknown from prior crystallographic structures of Pex5-cargo complexes
- (3) Pex5-Pex14_{NTD} interface extending beyond the canonical site

We do not make any claims beyond the above conclusions in our manuscript. The findings are supported by mutational analysis of the influence of identified interfaces on binding. Data supporting dynamics of Pex5-cargo interaction reported by others subsequently to our study in different organism (bioRxiv 2024.06.02.597006) suggests that dynamics may be common to the Pex5-cargo interface. These considerations were highlighted in the manuscript text in response to the first round of comments of the reviewers (R1) and are now strengthened in response to the above comment.

Regarding the claims pertaining to the influence (or lack of influence) of Pex14_{NTD} binding on Pex5-PTS1 interaction - in line with the reviewer suggestion, in the revised version of the manuscript we provide additional data demonstrating that the affinity of Pex5 for PTS1 peptide is moderately potentiated by Pex14 binding. This conclusion is consistent with the observation documented in this study that Pex14_{NTD} interacts directly with Pex5 TPR domain while binding at the proximal Wxxx(F/Y) motif.

2. *“Several suggestions that I think the authors could address with changes to the text:”*

a. *“Furthermore, our results show that the Pex14NTD binding at Pex5 proximal Wxxx(F/Y) site is not enough for structural reorganization of the Pex5 NTD and cargo release in vitro.” The reader is not given enough context in the introduction to understand the importance of this statement. It would be helpful to talk about domain architecture of Pex14, topology at the membrane, the need for release of cargo in the peroxisome in the intro. This is well covered in the discussion, but could be discussed more in the introduction.”*

Following the reviewer suggestion we have added the description of Pex14 architecture and topology in the introduction. The extended information supports the earlier present fragment on the postulated role of Pex14 in cargo release. We have also included additional data on the influence of Pex14_{NTD} on the Pex5-PTS1 affinity as described in more detail in response to comment 1 of this reviewer.

b. *“The density is present only at sites occupied by Pex5 and not empty PTS1 sites indicating it corresponds to a fragment of Pex5457-489 loop (Figure 3B; Supplementary Figure 12).” It seems possible this density could be a loop of MDH stabilized by the presence of Pex5 instead of a loop of Pex5 stabilized by MDH. Can the authors more clearly state why they think this density can definitively be assigned to PEX5 (ie rule out missing loops in MDH)?”*

The possibility that the density accounts for MDH loop stabilized in the presence of Pex5 is improbable as the MDH structure is well defined by electron density including almost all residues irrespective if Pex5 is present or absent at PTS1 sites. There is no “missing loop” (a fragment not accounted for by electron density) in apo-MDH which could rigidify upon interaction with Pex5 and that is why we conclude it is the flexible loop of Pex5 which rigidifies upon interaction with MDH.

c. *“The authors have chosen to exclude their previous ITC data of PEX5 and PEX14 interactions without PTS1 peptide present. This data could be useful for interpreting how PTS1 binding alters PEX5 affinity for PEX14, which is relevant to the authors discussion of PEX14 causing PEX5 to release its cargo.”*

ITC data of Pex5_{eTPR} (TPR domain containing the proximal WXXXF/Y motif) and Pex14_{NTD} interaction in the absence of PTS1 is included in the supplementary material and referenced in the revised manuscript.

d. *“The authors could discuss the conservation of the non-canonical interfaces for PEX14 NTD and PEX5.”*

The conservation of non-canonical interfaces of Pex14_{NTD} and Pex5 is now discussed in the revised version.

e. *“Proteins were purified by Ni-affinity (Ni-NTA) and size exclusion chromatography (SEC).” Authors should describe buffers, lysis, concentration methods for purification and EM grid preparation.”*

Detailed information is now provided in supplementary material under the title ‘supplementary method’.

Minor comments:

(i) *“Pex14 was additionally implemented in cargo extraction from the translocation pore.” Phrasing is off - did you mean implicated instead of implemented?”*

Yes. The spelling is now corrected in the revised version.

(ii) *“The maps accounting for the distal states of MDH-Pex5 complex (MP1-d and MP2-d) show additional density proceeding Pex5 residue Ser491 and facing MDH62-70 loop (Figure 3B; Supplementary Figure 12).” Did you mean preceding instead of proceeding?”*

Yes. This is now corrected in the revised version.

(iii) *“To furhter test the hypothesis, we mutated Pro490 residue” – typo in further”*

The typo is corrected in the revised version.

(iv) *“The density is not seen in a groove distal form Pex5 occupied site (red).” – typo in form”*

The typo has been corrected.

(v) *“Prior studies involving ATG and SCP2” – typo in AGT”*

The typo has been corrected in the revised version.

(vi) *“again demonstrating that the intractions at the intersubuint groove of MDH unfavorably affect the complex dynamics and affinity.” – typos in interactions and intersubunit”*

Both typos were corrected in the revised version.

Point-by-point responses to the reviewers' comments (R3)

Reviewer #1

Overall assessment:

"The changes in response to previous referee comments in the current revised version (NCOMMS-23-36304B) are appreciated. Overall, the manuscript has been improved."

Major comments:

1. *"I am still not really convinced by the concept of swinging motions, as swinging implies directivity, for which there is no evidence. At this point, I think this needs to be left for future discussion, when this paper is published."*

Following the reservation expressed by the reviewer and related suggestion of the editor to tone down the claims on conformational dynamics, we have removed the terms "swinging motion" and "conformational dynamics" from the revised manuscript. We now use the term "conformational heterogeneity" to refer to the variability observed in the MDH/Pex5/Pex14^{NTD} complex as generally accepted in the field (Punjani, A. and Fleet, D.J. 3D Variability Analysis: Resolving continuous flexibility and discrete heterogeneity from single particle cryo-EM images. Journal of Structural Biology, Volume 213, Issue 2, 2021).

Additional comments:

2. *"Statistical significance tests are now been mentioned in the text but are still not systematically applied to estimate the significance of the quantitative measurements in Tables 1 and 2. Mentioning the statistical significance in the text narrative is fine but cannot replace a systematic analysis where quantitative data are shown, i.e. Tables 1 and 2."*

Results of statistical significance test (p-values) have been incorporated in the Tables 1 and 2 to facilitate easier comparisons of reported values.

3. Referring to comment 2 above: *"This is of particular importance, as surprisingly a Pex5 version with a GSGS insert shows an approximately threefold improvement in binding to MDH. The statement "The change in affinity was entropy driven indicating that the secondary interactions have overall negative effect on affinity by restricting the conformational dynamics of the wild type interface", implying that it relieves the restriction on the conformational dynamics of wt Pex5 remains unproven by any complementary experiments. The next sentence in the text "the affinity of MDH(GSGS) is comparable to that of a short peptide encompassing PTS1 of MDH (residues 319-ARSKL-323; p>0.1) (Table 1) indicating that the introduction of the GSGS linker has indeed abolished all secondary interactions at the Pex5-MDH interface" even implies that the presence of the second site in the wild-type context has a negative effect on overall binding. So what is the functional impact of this second site? This remains an open question."*

The interpretation of entropy driven change in affinity of GSGS variant was toned down by changing the verb "indicating" into "suggesting", as indeed the conclusion is not proven by complementary evidence other than described in the paragraph referred to by the reviewer.

Identical change was introduced to the second sentence cited by the reviewer (on comparison of affinity of MDH(GSGS) and PTS1).

With the mutagenesis we were asking if the secondary sites contribute to the affinity of Pex5 and MDH. In other words, if the Pex5-MDH interaction is autonomous or non-autonomous as defined earlier by others and detailed in the first paragraph of the discussion. Our results suggest the autonomous-like model modulated by secondary interactions which have moderate negative effect on affinity and possibly affect the conformational heterogeneity of the complex as discussed in the “Discussion” section.

4. *Please remove the term "noncanonical" from the abstract, as the term will not be defined later on in the manuscript.*

The term “noncanonical” was removed from the abstract.

5. *“Unfortunately, the revised nomenclature for Pex5 constructs is still quite confusing. I could not find a definition for the acronym Pex5eTPR. Does the "e" stand for "extended"? According to Supplementary Table 4 Pex5hTPR "Refers to residues within full length Pex5 defined by density in the ternary MDH-Pex5-Pex14NTD complex structure", so the construct behind it is actually full-length Pex5. So, it's full-length Pex5 that was used for experiments, with limited structural visibility to Pex5hTPR. A statement on p. 10 about Pex5eTPR as "a construct roughly equivalent to Pex5hTPR" adds to further confusion. Please make a clear distinction between expressed and purified constructs, and expressing visibilities in structures.”*

(i) Pex5_eTPR was defined in Supplementary Table 4. The table was cited in the main text in the Materials and Methods section, but was not cited at the first appearance of the abbreviation. This is corrected in the revised version. “e” stands for “extended” as now defined in Supplementary Table 4.

(ii) Pex5_hTPR was not a separate construct, but an acronym referring to the residues defined by density in the structure reported. This abbreviation was removed in the revised version to avoid confusion with expressed and purified constructs. Visibilities in structures are described by defining the residue ranges in the revised version.

(iii) There are only two Pex5 constructs used in the entire study. Unless directly indicated otherwise in the manuscript text, the text full length Pex5 was used for structural studies and is referred to as Pex5. Truncated Pex5 was used for affinity studies and is referred to as Pex5_eTPR (extended TPR domain). All the mutants were prepared in Pex5_eTPR construct. No other acronyms are used in the revised version. If particular domains are discussed, they are referred to as such (eg. TPR domain; N-terminal domain).

6. *“I have noted that the new version of the manuscript contains various new changes that were not requested (or requested by the other referee, at least to some extent). In going through these changes, I have noticed that some new unclear statements have been introduced. At this point, I believe it would be beyond the role of this referee to comment on them in detail, to avoid an unintended role as advisor rather than referee. However, I feel it is important to flag this up to ensure that the final version to be eventually published is free of major deficiencies, ambiguities and/or errors.*

Just one example, to illustrate of the problem: Supplementary Figure 18 contains an acronym KdAV. I could not find an explanation for it. The measured values have a dimension, nM^{-1} I think. However, no dimension is given. These things needs to be corrected.”*

Introduced changes were made either directly in response to comments of one of the reviewers or as a secondary result of requested changes where the manuscript required additional adjustment to primary changes.

Data shown in Supplementary Figure 18 was introduced in response to one of the comments in R2 round of review (see “Overall assessment” section below where reviewer #2 refers to data shown in Supplementary Figure 18). K_{dAV} is an average K_d from free independent measurements. The acronym is now defined in the figure legend. Dimension of the value is provided in the revised version of the manuscript.

Reviewer #2

Overall assessment:

“The reviewers have addressed my concerns, and I appreciate their added fluorescence anisotropy experiments to assess how PEX14 NTD alters PEX5:PTS1 affinity.”

Additional comments:

1. *“The Pex5hTPR is used in the abstract and intro, but isn’t clearly defined until the results section. Perhaps in the other locations hTPR could be replaced by the residue numbers or hTPR should be defined earlier.”*

hTPR was removed altogether in the revised version as per suggestion of reviewer #1 and to address the concern of this reviewer. Constructs are now defined as specified in Supplementary Table 4 and regions defined by density in determined structures are described as residue numbers in the revised version. (See also response to comment 5 of Reviewer #1)

2. *“Supplementary Figure 7 – coloring the shown sidechains by atom (ie red = oxygen) would help clarify the H-bond network.”*

The requested coloring scheme was introduced in the revised figure to increase clarity.

3. *“Figure 2 legend – typo in “confirmations” which should be conformations.”*

The specified typo was corrected.

4. *“In Results section: “Pex5eTPR (residues 314-666), a construct roughly equivalent to Pex5hTPR”
- Please define hTPR by residue numbers here
- eTPR notation doesn’t match Supplementary Table 2: Pex5eTPR (residue 316-666)”*

hTPR acronym was removed altogether in the revised version (see response to comment 1 by this reviewer). The residue numbering of eTPR in Supplementary Table 1 was corrected (Supplementary table 2 does not contain reference to eTPR residue numbering).

5. *“Authors state: “To disrupt interface 1, we replaced the MDH62-70 loop with GS-linker. This resulted in a moderate decrease (~2.3-fold) ($p < 0.001$) in affinity towards Pex5eTPR indicating little contribution of the interface to affinity.” Please state the K_d of the GS linker mutant here or indicate that the MDH62-70 loop with GS-linker is (Delta62-70) in Table 1. When I look at Table 1, it is not immediately clear if the authors are referring to the mutant GSGS or Delta62-70.”*

The nomenclature referring to MDH variant with 62-70 loop replaced with GS-linker was made consistent in the main text and Table 1.

6. *“The authors discuss the comparison of the affinity between Pex5 and Pex14 NTD in the presence and absence of the PTS1 peptide in the rebuttal letter (“Affinity of Pex14NTD for Pex5eTPR determined by ITC in the presence of PTS1 peptide ($K_d = 53 \pm 12$) is lower ($p < 0.001$) compared to affinity determined in the absence of PTS1 peptide ($K_d = 35 \pm 5$). The statistical significance was determined using the Welch’s t -test.”), but it isn’t discussed in the manuscript (that I could find). I personally find it interesting that the presence of the PTS1 peptide increases the K_d between Pex5eTPR and Pex14 NTD, and would encourage the authors to include the statement from their rebuttal in the manuscript. It is interesting that N353E and Y358A Pex5 mutants do not have the same change in K_d with the addition of PTS1, suggesting that these mutants might decouple PTS1 binding and PEX14 NTD binding.”*

The relevant statements discussing the effect of PTS1 peptide on the affinity of Pex14_{NTD} for Pex5_{eTPR} are included in the revised manuscript according to the reviewer recommendation.